# Does Reinforcement Learning Really Incentivize Reasoning Capacity in LLMs Beyond the Base Model?

**Yang Yue** [1][*][†]  **Zhiqi Chen** [1][*]  **Rui Lu** [1]  **Andrew Zhao** [1]  **Zhaokai Wang** [2]  **Yang Yue** [1]
**Shiji Song** [1]  **Gao Huang** [1][✉]

[1] LeapLab, Tsinghua University       [2] Shanghai Jiao Tong University
*{le-y22, zq-chen23}@mails.tsinghua.edu.cn     {gaohuang}@tsinghua.edu.cn*

https://limit-of-rlvr.github.io

## Abstract

Reinforcement Learning with Verifiable Rewards (RLVR) has recently demonstrated notable success in enhancing the reasoning performance of large language models (LLMs), particularly in mathematics and programming tasks. It is widely believed that, similar to how traditional RL helps agents to explore and learn new strategies, RLVR enables LLMs to continuously self-improve, thus acquiring novel reasoning abilities that exceed the capacity of the corresponding base models. In this study, we take a critical look at *the current state of RLVR* by systematically probing the reasoning capability boundaries of RLVR-trained LLMs across various model families, RL algorithms, and math/coding/visual reasoning benchmarks, using pass@$k$ at large $k$ values as the evaluation metric. While RLVR improves sampling efficiency towards correct paths, we surprisingly find that current training does *not* elicit fundamentally new reasoning patterns. We observe that while RLVR-trained models outperform their base models at smaller values of $k$ (*e.g.*, $k$=1), base models achieve higher pass@$k$ score when $k$ is large. Moreover, we observe that the reasoning capability boundary of LLMs often narrows as RLVR training progresses. Further coverage and perplexity analysis shows that the reasoning paths generated by RLVR models are already included in the base models' sampling distribution, suggesting that their reasoning abilities originate from and are *bounded* by the base model. From this perspective, treating the base model as an upper bound, our quantitative analysis shows that six popular RLVR algorithms perform similarly and remain far from optimal in fully leveraging the potential of the base model. In contrast, we find that distillation can introduce new reasoning patterns from the teacher and genuinely expand the model's reasoning capabilities. Taken together, our findings suggest that current RLVR methods have not fully realized the potential of RL to elicit genuinely novel reasoning abilities in LLMs. This underscores the need for improved RL paradigms–such as continual scaling and multi-turn agent-environment interaction–to unlock this potential.

## 1 Introduction

The development of reasoning-centric large language models (LLMs), such as OpenAI-o1 [1], DeepSeek-R1 [2], and Kimi-1.5 [3], has significantly advanced the frontier of LLM capabilities, particularly in solving complex logical tasks involving mathematics and programming. In contrast to traditional instruction-tuned approaches that rely on human-curated annotations [4, 5], the key driver

---

[*] Equal Contribution. [†] Project Lead. [✉] Corresponding Author. The first author Yang Yue (乐洋) and the sixth author Yang Yue (乐阳) share the same English name but different Chinese names.

39th Conference on Neural Information Processing Systems (NeurIPS 2025).

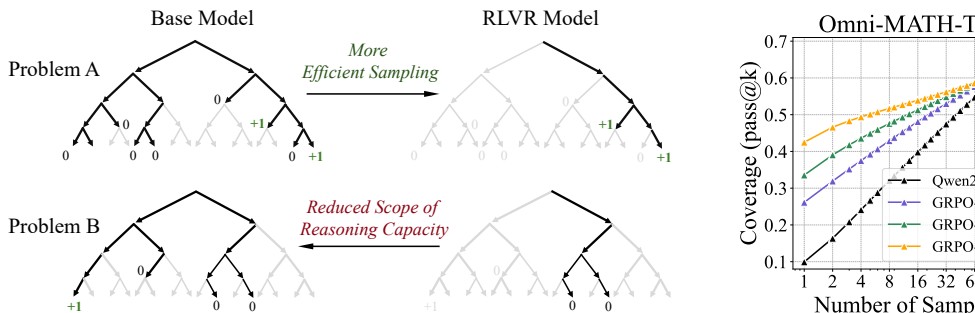

Figure 1: **(Left)** The effect of current RLVR on LLM's reasoning ability. Search trees are generated by repeated sampling from the base and RLVR-trained models for a given problem. Grey indicates paths that are unlikely to be sampled by the model, while **black** indicates paths that are likely to be sampled. Green indicates correct paths, which has positive rewards. Our key finding is that all reasoning paths in the RLVR model are already present in the base model. For certain problems like Problem A, RLVR training biases the distribution toward rewarded paths, improving sampling efficiency. However, this comes at the cost of reduced scope of reasoning capacity: For other problems like Problem B, the base model contains the correct path, whereas that of the RLVR model does not. **(Right)** As RLVR training progresses, the average performance (*i.e.*, pass@1) improves, but the coverage of solvable problems (*i.e.*, pass@256) decreases, indicating a reduction in LLM's reasoning boundary.

behind this leap forward is large-scale *Reinforcement Learning with Verifiable Rewards* (RLVR) [6, 2]. RLVR starts with a pretrained base model or one fine-tuned on long chains of thought (CoT) data, optimizing it via reinforcement learning based on simple, automatically computable rewards. These rewards are determined by whether the model's output matches a ground-truth solution in mathematics or passes unit tests in code, thus enabling scalability without human labeling. This framework has gained significant attention due to its simplicity and practical effectiveness. In traditional RL settings such as game playing (*e.g.*, Atari, Go), agents often autonomously discover new strategies and surpass even human-level performance through self-improvement [7, 8]. Inspired by this success, it is widely believed that RLVR similarly enables LLMs to autonomously develop novel reasoning patterns, including enumeration, self-reflection, and iterative refinement, surpassing the capabilities of their base models [2]. Consequently, RLVR has been considered a promising path toward continuously self-evolving LLMs, potentially bringing us closer to more powerful intelligence [2].

However, despite its empirical success, the underlying effectiveness of current RLVR remains underexamined. This raises a fundamental question: ***Does current RLVR genuinely enable LLMs to acquire novel reasoning abilities–similar to how traditional RL discovers new strategies through exploration–or does it simply utilize reasoning patterns already in the base model?***

To rigorously answer this question, we must first assess the reasoning capability boundaries of both base and RLVR-trained models. Traditional evaluation metrics rely on average score from greedy decoding or nucleus sampling [9], which reflects average-case behavior. However, these metrics risk underestimating the true potential of a model, especially if it fails on difficult problems after limited attempts, despite being capable of solving them with more sampling. To overcome this limitation, we adopt the pass@k metric [10], where a problem is considered solved if any of the k sampled outputs is correct. By allowing multiple attempts, pass@k reveals whether a model has the potential to solve a problem. The average pass@k across a dataset thus reflects the proportion of problems a model can potentially solve within k trials, offering a more robust view of its reasoning boundary. This provides a rigorous test on whether the RLVR training yields fundamentally transcending capacity, enabling the model to solve problems that the base model cannot.

Using the pass@k metric, we conduct extensive experiments across various benchmarks, covering multiple LLM families, model sizes, and RLVR algorithms to compare base models with their RLVR-trained counterparts. We uncover several surprising findings that offer a more comprehensive assessment of the effectiveness of current RLVR training and reveal the gap between existing RLVR methods and the ideal goals of RL-discovering genuinely new reasoning strategies:

• **Current RLVR models exhibit narrower reasoning coverage than their base models.** In pass@k curves, although RLVR models outperform their base models at small k, it is surprising that base models consistently surpass RLVR models across all benchmarks and LLM families as k increases. This suggests that current RLVR training does *not* expand, and even reduce the scope of reasoning over solvable problems. Manual inspection of model responses shows that, for most

problems, the base model can produce *at least one* correct CoT, implying that it can already generate correct reasoning paths for problems that were previously considered only solvable for RLVR models.

• **Reasoning paths generated by current RLVR model already exist in its base model.** To further investigate this phenomenon, we analyze the accuracy distribution. The results show that although RLVR improves average performance (*i.e.*, pass@1) by sampling more efficiently on problems already solvable by the base model, it does not enable the model to solve new problems. Further perplexity analysis reveals that the reasoning paths produced by RLVR models already exist within the output distribution of the base model. These findings indicate that RLVR does not introduce fundamentally new reasoning capabilities and that the reasoning capacity of current RLVR models remains bounded by that of its base model. This effect of RLVR is illustrated in Figure 1 (left).

• **Current RLVR algorithms perform similarly and remain far from optimal.** Treating the base model as an upper bound, we define the *sampling efficiency gap* ($\Delta_{\text{SE}}$), shown in Figure 8 (top), as the difference between an RL model's pass@1 and the base model's pass@$k$ (with $k = 256$ as a proxy for upper-bound performance). This metric quantifies how closely an RL algorithm approaches the optimal bound. Across all algorithms (e.g., PPO, GRPO, Reinforce++), $\Delta_{\text{SE}}$ shows only minor variation yet remains consistently large, suggesting that current RLVR methods, while improving sampling efficiency, are still far from optimal.

• **RLVR and distillation are fundamentally different.** While RLVR improves reasoning scores by more efficiently sampling high-reward outputs, it does not elicit new reasoning capabilities and remains constrained within the base model's capacity. In contrast, distillation can transfer new reasoning patterns from a stronger teacher to the student. As a result, distilled models often demonstrate an expanded reasoning scope beyond that of the base model.

In conclusion, our findings show that current RLVR methods, while improving sampling efficiency, do not elicit novel reasoning beyond the base model's capabilities. This highlights a gap between existing RLVR methods and the goals of reinforcement learning, underscoring the need for improved RL paradigms such as continual scaling, better exploration, and multi-turn agent interaction.

## 2 Preliminaries

In this section, we first outline the fundamentals of RLVR, then introduce the pass@$k$ metric to evaluate reasoning boundaries, and explain why it is preferred over alternatives like best-of-$N$.

### 2.1 Reinforcement Learning with Verifiable Rewards

**Verifiable Rewards.** Let $\pi_\theta$ be an LLM with parameters $\theta$ that generates a token sequence $\mathbf{y} = (y_1, \ldots, y_T)$ conditioned on a natural-language prompt $x$. A deterministic *verifier* $\mathcal{V}$ returns a binary reward: $r = \mathcal{V}(x, \mathbf{y}) \in \{0, 1\}$, where $r = 1$ if and only if the model's final answer is exactly correct. A format reward may also be added to encourage the model to explicitly separate the reasoning process from the final answer. The goal of RL is to learn a policy to maximize the expected reward: $J(\theta) = \mathbb{E}_{x \sim \mathcal{D}} \left[ \mathbb{E}_{\mathbf{y} \sim \pi_\theta(\cdot|x)}[r] \right]$, where $\mathcal{D}$ is the distribution of prompts.

**RLVR Algorithms.** Proximal Policy Optimization (PPO) [11] proposed using the following clipped surrogate to maximize the objective:
$$\mathcal{L}_{\text{CLIP}} = \mathbb{E}\left[\min(r_t(\theta)A_t, \ \text{clip}(r_t(\theta), 1 - \epsilon, 1 + \epsilon)A_t)\right], \tag{1}$$
where $r_t(\theta) = \pi_\theta(y_t|x, \mathbf{y}_{<t})/\pi_{\theta_{\text{old}}}(y_t|x, \mathbf{y}_{<t})$, and $A_t$ is the advantage estimated by a value network $V_\phi$. KL divergence term is optionally applied, to constrain the model from deviating too far from the original policy. More algorithms are introduced in Section D.5.

**Policy Gradient.** PPO and its variants belong to the policy gradient class of RL [12, 13]. These methods learn exclusively from *on-policy samples*, *i.e.*, samples generated by the current LLM. In the context of verifiable rewards, the training objective generally *maximizes the log-likelihood of samples with correct answers and minimizes the likelihood of those with incorrect answers.*

**Zero RL Training** applies RL directly to the base model without any supervised fine-tuning (SFT) [2]. To clearly study the effect of RLVR, we follow this zero-RL setting for all math tasks using pretrained models as start model. However, for coding and visual reasoning tasks, open-source work typically uses instruction-tuned models as starting points, primarily due to the training instability and limited effectiveness of using a pure zero-RL setting. Following this convention, we compare the finetuned model with its RLVR-trained counterpart to focus solely on the effect of RLVR.

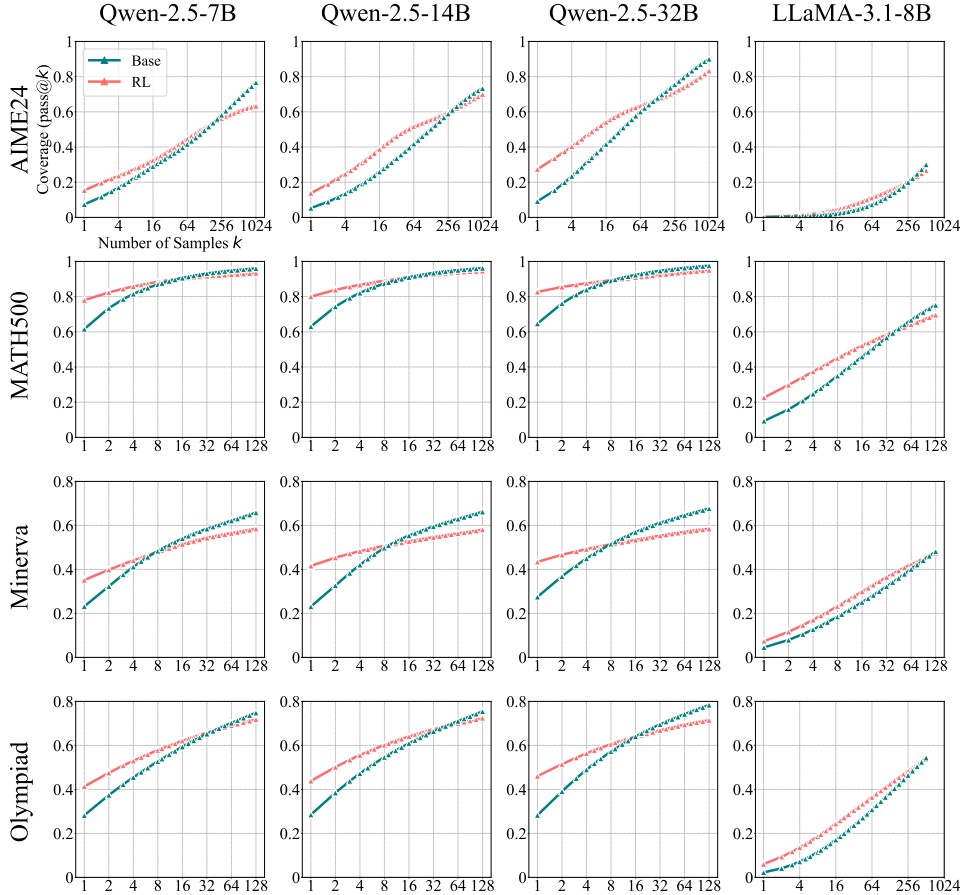

Figure 2: Pass@$k$ curves of base models and their RLVR-trained counterparts across multiple mathematical benchmarks. When $k$ is small, RL-trained models outperform their base versions. However, as $k$ increases to the tens or hundreds, base models consistently catch up and surpass RL-trained models. More results on GSM8K and AMC23 can be found at Figure 10.

## 2.2 Metrics for LLM Reasoning Capacity Boundary

**Pass@$k$ Metrics.** Accurately measuring the reasoning ability boundary of base and RL models is challenging, as methods like greedy decoding or the average of nucleus samplings [9] only reflect average-case performance. To accurately measure the reasoning ability boundary, we extend the commonly used pass@$k$ metric from code generation [14] to all tasks with verifiable rewards. Given a problem, we sample $k$ outputs from the model. The pass@$k$ value for this question is 1 if at least one of the $k$ samples passes verification; otherwise, it is 0. The average pass@$k$ value over the dataset reflects the proportion of problems in the dataset that the model can solve within $k$ trials, providing a rigorous evaluation of the reasoning capacity coverage of LLMs. We adopt an unbiased, low-variance estimator for computing to calculate pass@$k$, as detailed in Section A.2.

**Comparison with Best-of-$N$ and Majority Voting.** Best-of-$N$ [15] and majority voting are practical methods for selecting correct answers, but they may overlook a model's full reasoning potential. In contrast, we use pass@$k$ *not to assess practical utility* but to investigate the boundaries of reasoning capacity. If a model produces a correct solution in any of the $k$ samples, we treat the problem as within its potential scope. Thus, if RL enhances reasoning, the RL-trained model should succeed in more such problems than the base model. Methods like Best-of-$N$ or majority voting may miss these successes if the correct answer is not selected by the verifier or voting.

**Random Guessing Issue**. For *coding* tasks, where a compiler and predefined unit test cases are used as verifiers, the pass@$k$ value can accurately reflect whether the model can solve the problem. In *mathematics*, the issue of "guessing" can become pronounced as $k$ increases, where a model may generate an incorrect CoT but still accidentally arrive at the correct answer. To address this, we manually check the correctness of CoT for a subset of model outputs as detailed in Section 3.1. By combining reuslts on math with manually checking and coding, we rigorously evaluate the scope

of LLM's reasoning capacity. Another caveat is that, with an astronomically large $k$, even uniform sampling over the token dictionary would stumble upon the correct reasoning path–though this is infeasible within today's time and compute budgets. Crucially, we find that the base model already produces correct outputs at realistic values ($k = 128$ or $1024$), well within practical resource limits.

Table 1: Experimental setup for assessing RLVR's effect on the reasoning boundaries of LLMs.

| Task | Start Model | RL Framework | RL Algorithm(s) | Benchmark(s) |
|---|---|---|---|---|
| **Mathematics** | LLaMA-3.1-8B Qwen2.5-7B/14B/32B-Base Qwen2.5-Math-7B | SimpleRLZoo Oat-Zero DAPO | GRPO | GSM8K, MATH500 Minerva, Olympiad AIME24, AMC23 |
| **Code Generation** | Qwen2.5-7B-Instruct DeepSeek-R1-Distill-Qwen-14B | Code-R1 DeepCoder | GRPO | LiveCodeBench HumanEval+ |
| **Visual Reasoning** | Qwen2.5-VL-7B | EasyR1 | GRPO | MathVista MathVision |
| **Deep Analysis** | Qwen2.5-7B-Base Qwen2.5-7B-Instruct DeepSeek-R1-Distill-Qwen-7B | VeRL | PPO, GRPO Reinforce++ RLOO, ReMax, DAPO | Omni-Math-Rule MATH500 |

# 3 RLVR's Effect on Reasoning Capacity Boundary

With the evaluation metrics for reasoning boundaries established, we now conduct a comprehensive evaluation of the base and RLVR models through extensive experiments. Our analysis is organized by task category, covering three representative domains: mathematics, code generation, and visual reasoning. The overall experimental setup is summarized in Table 1.

**Evaluation Protocol.** For sampling procedures for both base and RLVR models, we use a temperature of 0.6 and a top-$p$ value of 0.95, allowing a maximum generation of 16,384 tokens. We also show the effect of different temperature settings in Figure 17. For evaluation of the base model, a common practice is to include few-shot examples in the prompt to guide the output [5, 16, 17]. However, to ensure a fair and unbiased comparison, we deliberately avoid using few-shot prompts for base models, eliminating any potential confounding effects on reasoning that might be introduced by in-context examples. For evaluating both the base and RLVR models, we use the same zero-shot prompt as in RLVR training, or the default prompt provided by the benchmark, ensuring a consistent setup across both models. Interestingly, although base models often produce unformatted or non-sensical responses without few-shot guidance, we observe that with sufficient sampling, they are still capable of generating correctly formatted outputs and successfully solving complex problems. Prompt templates for training and evaluation are provided in Section E.

## 3.1 RLVR for Mathematical Reasoning

**Models and Benchmarks.** In math problems, models are required to generate a reasoning process (*i.e.*, CoT) along with the final answer. To ensure the robustness of conclusions, we experiment with multiple LLM families, primarily Qwen2.5 (7B/14B/32B base variants) [16] and additionally LLaMA-3.1-8B [5]. We adopt RLVR models released by SimpleRLZoo [18], which train zero-RL models using GRPO on GSM8K and the MATH training set, with correctness reward only, excluding any format-based reward. We compare the pass@$k$ curves of base and zero-RL models on benchmarks of varying difficulty: GSM8K [15], MATH500 [19], Minerva [20], Olympiad [21], AIME24, and AMC23. Additionally, we include the RLVR model Oat-Zero-7B and DAPO-32B [22, 23]. These two models are characterized by strong performance on the challenging AIME24 benchmark.

**The Effect of RLVR: Increased Likelihood of Correct Samples, Decreased Coverage of Solvable Problems.** As shown in Figure 2, we consistently observe a contrasting trend between small and large $k$ values. When $k$ is small (*e.g.*, $k = 1$, equivalent to average-case accuracy), RL-trained models outperform their base counterparts. This aligns with the common observation that RL improves performance, suggesting that RLVR makes models significantly more likely to sample correct responses. However, as $k$ increases, with steeper curves, base models consistently catch up to and eventually surpass RL-trained models across all benchmarks, indicating their broader coverage of solvable problems. For example, on the Minerva benchmark with a 32B-sized model, the base model outperforms the RL-trained model by approximately 9% at $k = 128$, implying that it can solve around 9% more problems in the validation set.

We further examine RL models trained with Oat-Zero and DAPO. As shown in Figure 11, although the RL model initially demonstrates a strong performance, nearly 30% higher than the base model, it is eventually surpassed by the base model. Based on these results, we conclude that RLVR increases the likelihood of sampling correct responses at low $k$, but narrows the model's overall coverage. We further analyze the root cause of this phenomenon in Section 4.1.

**CoT Case Analysis.** We present the correct CoTs sampled from the base model in Figure 20 and Figure 21, manually selected from 2048 samplings for the hardest questions in AIME24. The responses from the base model tend to be long CoTs and exhibit reflective behavior, highlighting the strong reasoning ability inherent in the base model.

**Validity of Chain-of-Thought.** For mathematical problems, the common evaluation is based solely on the correctness of the final answer, with the risk of "hacking". To accurately reflect the reasoning ability boundary using pass@$k$, it is important to assess how many solved problems result from sampling genuinely correct CoTs, rather than from lucky guesses. Following [10], we manually inspect all CoTs that led to correct answers to the most challenging solvable problems in the GSM8k dataset – those with an average accuracy below 5% but above 0%. The base model answered 25 such questions, with 24 containing *at least one* correct CoT. Similarly, the RL-trained model answered 25 questions, 23 of which included *at least one* correct CoT. We also manually check the CoTs for problems in the challenging AIME24 benchmark with an average accuracy below 5%. Details can be found in Section D.2. The base model answered 7 such questions, with 5 out of 6 containing *at least one* correct CoT (excluding one ambiguous case of correctness due to skipped reasoning steps). Similarly, the RL-trained model answered 6 questions, 4 of which included *at least one* correct CoT. These results suggest that the base model can sample valid reasoning paths to solve the problems.

## 3.2 RLVR for Code Generation

**Models and Benchmarks.** We adopt the open-sourced RLVR-trained model, CodeR1-Zero-Qwen2.5-7B [24], which trains zero-RL models on 12K LeetCode and TACO samples over 832 steps, based on Qwen2.5-7B-Instruct-1M [25]. For evaluation, models are assessed on LiveCodeBench v5, comprising 279 problems that span from August 2024 to January 2025 [26], as well as HumanEval+ and MBPP+ [27]. We also evaluate the most powerful open-source RLVR-trained coding LLM, DeepCoder-14B [28], built on DeepSeek-R1-Distill-Qwen-14B. Here both models take 32k response length. Due to their high computational cost, we evaluate them only on LiveCodeBench as a representative benchmark.

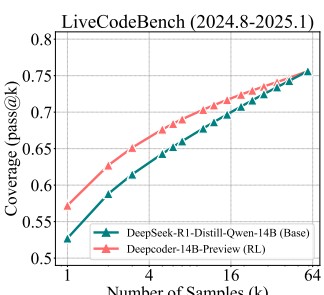

Figure 3: RLVR for Coding.

**The Effect of RLVR.** Since passing all unit tests is nearly impossible to achieve by guesswork, pass@$k$ provides a reliable measure of a model's reasoning boundary. As shown in Figure 3, Figure 12, and Figure 4 (left), the effects of RLVR on three code generation benchmarks exhibit trends that are highly consistent with those observed in mathematical benchmarks.

## 3.3 RLVR for Visual Reasoning

**Models and Benchmarks.** In visual reasoning, models must jointly interpret visual and textual inputs to solve complex reasoning problems. This has gained significant attention in the multimodal community since the rise of LLM reasoning [29–31]. For our experiments, we select math within visual contexts as a representative task. We use the EasyR1 framework [31] to train Qwen2.5-VL-7B [32] on Geometry3K [33], and evaluate its visual reasoning capabilities on filtered MathVista-TestMini [34] and MathVision-TestMini [35], where multiple-choice questions are removed.

**The Effect of RLVR.** As shown in Figure 4 (right), the effects of RLVR on visual reasoning are highly consistent with those observed in math and coding benchmarks. This suggests that the original model has broader coverage of solvable questions even in multimodal tasks.

**Validity of Chain-of-Thought.** Similarly, we manually inspect a subset of the most challenging problems, *i.e.* those with an average accuracy below 5%. We find that for both the original and RL models, 7 out of 8 problems have *at least one* correct CoT. These results support the validity of CoTs.

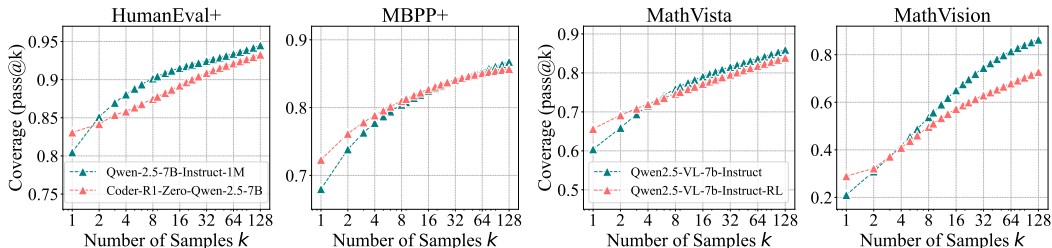

Figure 4: Pass@$k$ curves of base and RLVR models. **(Left)** Code Generation. **(Right)** Visual Reasoning.

# 4 Deep Analysis

In this section, we conduct a deeper analysis of the effects of current RLVR training. We also highlight the distinct characteristics of distillation in comparison to RLVR. In addition, we design controlled experiments to examine the impact of different RL algorithms and design choices.

## 4.1 Reasoning Paths Already Present in Base Models

**Accuracy Distribution Analysis.** Experiments in Section 3 reveal a surprising trend: the base model covers a wider range of solvable problems than the RLVR-trained model. To better understand this, we analyze how the accuracy distribution changes before and after RLVR training. As shown in Figure 5, RLVR increases the frequency of high accuracies near 1.0 and reduces the frequency of low accuracies (*e.g.*, 0.1, 0.2). However, a deviation from this trend is the *increased frequency at accuracy 0* — indicating that RLVR leads to more unsolvable problems. This also explains the improvement of RLVR in average scores, driven not by solving new problems but by improving sampling efficiency on problems already solvable by the base model. Additional accuracy histograms are provided in Figure 14.

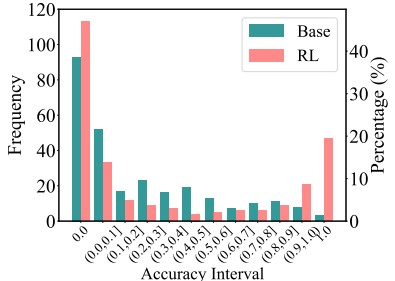

Figure 5: Qwen2.5-7B Accuracy Histogram on Minerva.

**Solvable-Problem Coverage Analysis.** To further investigate, we compare the set of solvable questions for both the base model and its corresponding RL-trained version on AIME24 and MATH500. We find that there are many cases where the base model solves a problem but the RLVR model fails, and very few where RLVR succeeds while the base model does not, as shown in Table 2. Details can be found at Section D.7. As shown in Table 5, the set of problems solved by the RL-trained model is nearly a subset of those solvable by the base model. A similar trend is observed in coding tasks as shown in Table 6.

Table 2: We evaluate on AIME24 ($k = 1024$) and MATH500 ($k = 128$). The table reports the solvable/unsolvable fraction of problems falling into four categories.

| Base | SimpleRLZoo | AIME24 | MATH500 |
|:---:|:---:|:---:|:---:|
| ✓ | ✓ | 63.3% | 92.4% |
| ✓ | ✗ | 13.3% | 3.6% |
| ✗ | ✓ | 0.0% | 1.0% |
| ✗ | ✗ | 23.3% | 3.0% |

This raises the natural question: Do all reasoning paths generated by RL-trained models already exist within the output distribution of their base models?

**Perplexity Analysis**. To answer this question, we utilize the metric *perplexity*. Given a model $m$, a problem $x$, and a response $\mathbf{Y} = (y_1, \ldots, y_T)$ (can be generated by the same model, another model, or humans), the perplexity is defined as the exponentiated average negative log-likelihood of a sequence:

$$\text{PPL}_m(\mathbf{Y} \mid x) = \exp\left(-\frac{1}{T}\sum_{t=1}^{T} \log P(y_t \mid x, y_1, \ldots, y_{t-1})\right),$$

which reflects the model's ability to predict the given response $\mathbf{Y}$ conditioned on the prompt $x$. Lower perplexity indicates that the model has a higher likelihood of generating this response.

We randomly sample two problems from AIME24 and employ Qwen2.5-7B-Base and SimpleRL-Qwen2.5-7B-Base to generate 16 responses for each problem, denoted as $\mathbf{Y}_{\text{base}}$ and $\mathbf{Y}_{\text{RL}}$, respectively. We also let OpenAI-o1 [1] generate 8 responses, denoted as $\mathbf{Y}_{\text{GT}}$. As shown in Figure 6, the distribution of $\text{PPL}_{\text{Base}}(\mathbf{Y}_{\text{RL}}|x)$ closely matches the lower portion of the $\text{PPL}_{\text{Base}}(\mathbf{Y}_{\text{Base}}|x)$ distribution,

corresponding to responses that the base model tends to generate. This suggests that the responses from RL-trained models are highly likely to be generated by the base model. In Section D.4, we show that $\text{PPL}_{\text{Base}}(\mathbf{Y}_{\text{RL}}|x)$ gradually decreases as RL training progresses, indicating that RLVR mainly sharpens the distribution within the base model's prior rather than expanding beyond it.

**Summary.** Combining the above analyses, we arrive at three key observations. First, problems solved by the RLVR model are also solvable by the base model; the observed improvement in average scores stems from more efficient sampling on these already solvable problems, rather than learning to solve new problems. Second, after RLVR training, the model often exhibits narrower reasoning coverage compared to its base model. Third, all the reasoning paths exploited by the RLVR model are already present in the sampling distribution of the base model. These findings indicate that RLVR does not introduce fundamentally new reasoning capabilities and that the reasoning capacity of the trained model remains bounded by that of its base model.

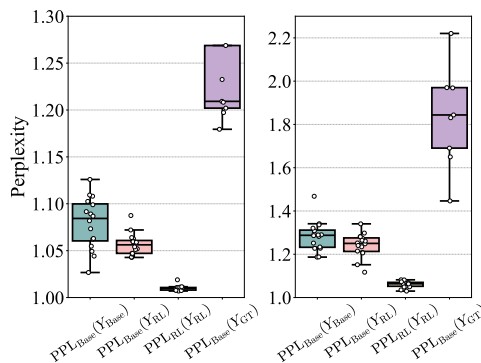

Figure 6: Perplexity distribution of responses. The conditioning problem $x$ is omitted in the figure.

## 4.2 Distillation Expands the Reasoning Boundary

In addition to direct RL training, another effective approach to improving the reasoning ability of small base models is distillation from a powerful reasoning model [2]. This process is analogous to instruction-following fine-tuning in post-training. However, instead of using short instruction-response pairs, the training data consist of long CoT reasoning traces generated by the teacher model. Given the limitations of current RLVR in expanding reasoning capabilities, it is natural to ask whether distillation exhibits similar behavior. We focus on a representative model, DeepSeek-R1-Distill-Qwen-7B, which distills DeepSeek-R1 into Qwen2.5-Math-7B. We compare it with the base model Qwen2.5-Math-7B and its RL-trained counterpart Qwen2.5-Math-7B-Oat-Zero and include Qwen2.5-Math-7B-Instruct as an additional baseline. As shown in Figure 7, the pass@$k$ curve of the distilled model is consistently and significantly above that of the base model. This indicates that, unlike RL that is fundamentally bounded by the reasoning capacity of the base model, distillation introduces new reasoning patterns learned from a stronger teacher model. As a result, the distilled model is capable of surpassing the reasoning boundary of the base model.

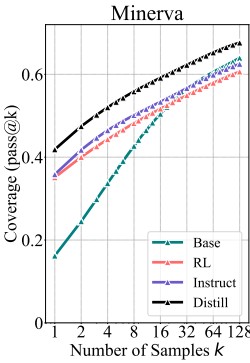

Figure 7: pass@$k$ of base, Instruct, RLVR, and distilled models.

## 4.3 Effects of Different RL Algorithms

As discussed previously, the primary effect of RL is to enhance sampling efficiency rather than to expand a model's reasoning capacity. To quantify this, we propose the *Sampling Efficiency Gap* ($\Delta_{\text{SE}}$), defined as the difference between the RL-trained model's pass@1 and the base model's pass@$k$ (we use $k = 256$ in our evaluation). Lower $\Delta_{\text{SE}}$ is better. Here we conduct clean experiments to study the effect of different RL algorithms in enhancing sampling efficiency.

**Experiment Setup.** We re-implement popular RL algorithms using the VeRL framework [36] for fair comparison, including PPO [11], GRPO [37], Reinforce++ [38], RLOO [39], ReMax [40], and DAPO [23]. Following DAPO [23] and Oat-Zero [22], we remove the KL term to avoid constraining model learning. During training, we use the AdamW optimizer [41] with a constant learning rate of $10^{-6}$. For rollout, we employ a prompt batch size of 256 and generate 8 responses per prompt. The maximum rollout length is set to 8,192 tokens, and the sampling temperature is set as 1.0. We use a PPO mini-batch size of 256.

To assess in-domain and out-of-domain generalization under RLVR, we split Omni-MATH-Rule, a subset of Omni-MATH [42] containing verifiable problems, into a training set (2,000 samples) and an in-domain test set (821 samples), and use MATH500 as the out-of-domain benchmark.

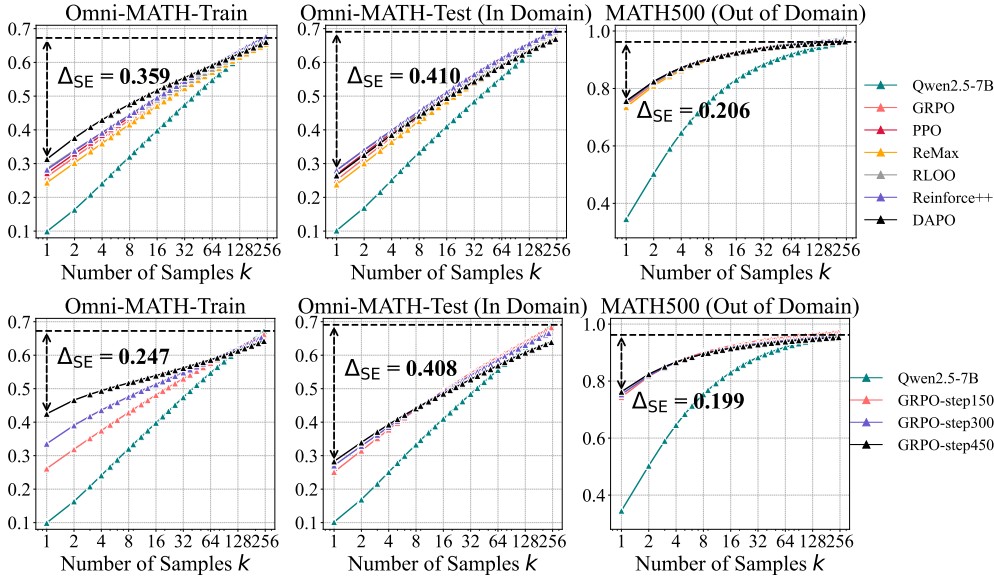

Figure 8: **(Top)** Different RL algorithms. **(Bottom)** Different RL training steps. The detailed values for each point at pass@1 and pass@256 are provided in Table 3 and Table 4.

**Results.** As shown in Figure 8 (top), although different RL algorithms exhibit slight variations in both pass@1 and pass@256, these differences are not fundamental. Different RL algorithms yield slightly different $\Delta_{\text{SE}}$ values (*i.e.*, ranging from GRPO's 43.9 to RLOO's best 42.6 on the in-domain test set). Furthermore, we observe that $\Delta_{\text{SE}}$ remains consistently above 40 points across different algorithms, highlighting that existing RL methods are still far from achieving optimal sampling efficiency. This suggests that novel RL algorithms or entirely new paradigms may be necessary to approach the upper bound. Additional observations can be found at Section D.5.

## 4.4 Effects of RL Training

**Asymptotic Effects.** Based on the setup in Section 4.3, we investigate the effect of the training steps on the asymptotic performance of the model. As shown in Figure 1 (right), as RL training progresses, pass@1 on the training set consistently improves from 26.1 to 42.5. However, as RLVR training progresses, pass@256 progressively decreases, indicating a reduced reasoning boundary.

**Effect of Number of Rollouts $n$.** The training hyperparameter $n$, the number of responses per prompt, can affect pass@$k$ by enabling broader exploration during training. We increase $n$ from 8 to 32. As shown in Figure 16, pass@$k$ improves slightly over $n = 8$, but the RL-trained model is still eventually outperformed by the base model. We leave the question of whether scaling RLVR training can eventually surpass the base model to future investigation.

**Effect of KL Loss.** To control model deviation, some prior work adds a KL penalty. We ablate this by applying a KL term with coefficient 0.001. As shown in Figure 16, the KL-regularized model achieves similar pass@1 to GRPO without KL, but with a much lower pass@128.

## 4.5 Effects of Entropy

As RL training progresses, the model's output entropy typically decreases [23], which may contribute to a reduced reasoning boundary due to less diverse output. To investigate this factor, we increase the generation temperature of the RLVR-trained model to match the output entropy of the base model at $T = 0.6$. As shown in Figure 18, although the RLVR model performs slightly better pass@$k$ at higher temperatures compared to its own performance at $T = 0.6$, it still underperforms the base model across pass@$k$. This suggests that while reduced entropy contributes to the narrowing of the reasoning boundary, it alone does not fully account for the reduction.

## 4.6 Effects of Model Size Scaling

Scaling plays a central role in the capabilities of contemporary LLMs. It remains an important question whether the conclusions drawn continue to hold as model size increases. For many large models, isolating the effect of RLVR is not feasible. For example, in the case of GPT-o1, the base model is not publicly accessible. Qwen3-235B [43] is trained through multiple stages, including RLVR and long-context CoT supervised fine-tuning, which makes it impossible to disentangle the impact of RLVR alone. For Deepseek-R1-Zero, the absence of a publicly hosted API forced us to self-host the model, but throughput was limited to around 50 tokens per second at a maximum sequence length of 32k, rendering pass@$k$ evaluation currently impractical. As a more tractable alternative, we selected the Magistral-Medium-2506 API to conduct a preliminary set of experiments. This model is trained using pure RL, with Mistral-Medium-3-2505 as the starting model [44]. Although the model size is not disclosed, Magistral-Medium performs comparably to Deepseek-R1 and is positioned near the frontier in terms of reasoning capability.

We queried the models using a maximum context length of 40k as the original paper does. Once again, we observed that RLVR provides significant gains at low $k$, but little or no improvement at higher $k$. Specifically, at $k = 1$, the RLVR-enhanced model solves approximately 7 more problems on AIME24 and 8 more on AIME25 compared to its base version. However, as $k$ increases, the performance gap steadily narrows. These observations suggest that our conclusion continues to hold even for current, highly capable, near-frontier reasoning models. Whether this

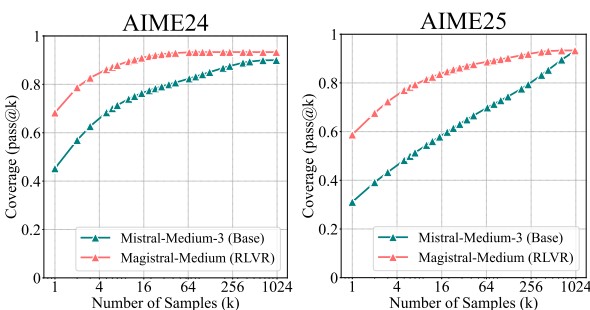

Figure 9: pass@$k$ curves of Magistral-Medium.

trend persists as more compute, such as pre-training scale budgets, is dedicated to RL training remains a critical question for the future of LLM reasoning.

## 5 Related Work

We summarize key related works on the analysis of RLVR here and provide a more comprehensive discussion in Appendix B. While recent RLVR methods have achieved impressive empirical results [2, 6], their fundamental impact on reasoning remains underexplored. Several studies [45–47] suggest that reflective behaviors in RLVR models originate from the base models rather than being learned through reinforcement learning. Dang *et al*. [48] observed a decline in pass@$k$ performance post-RLVR training, but their analysis was limited in scope. More importantly, they did not explore the relationship between the base model and the RL model. Deepseek-Math [37] also observed similar trends, but their study was limited to a single instruction-tuned model and two math benchmarks. In contrast, our work systematically investigates a wide range of models, tasks, and RL algorithms to accurately assess the effects of current RLVR methods and models. We further provide in-depth analyses, including accuracy distributions, reasoning coverage, perplexity trends, and comparison against distilled models, offering a comprehensive understanding of RLVR's capabilities and limitations.

## 6 Conclusion and Discussion

RLVR is widely regarded as a promising approach to enable LLMs to continuously self-improve and acquire novel reasoning capabilities. In this paper, we systematically investigate the effect of current RLVR methods on the reasoning capacity boundaries of LLMs. Surprisingly, our findings reveal that current RLVR does not elicit fundamentally new reasoning patterns; instead, the reasoning capabilities of RLVR-trained models remain bounded by those of their base models. These results indicate that current RLVR methods have not fully realized the potential of reinforcement learning to elicit novel reasoning abilities in LLMs through exploration and exploitation. This limitation may stem from the lack of effective exploration strategies in the vast language space or the absence of multi-turn agent-environment interactions needed to generate novel experience. We provide a more detailed discussion of the possible causes of this gap and promising future directions in Section C.

## Acknowledgements

This work is supported in part by the National Key R&D Program of China under Grant 2022ZD0114903, the National Natural Science Foundation of China under Grants 42327901 and U24B20173, and the Scientific Research Innovation Capability Support Project for Young Faculty under Grant ZYGXQNJSKYCXNLZCXM-I20.

## Author Contributions

**All authors** made valuable contributions to the experimental design, analysis, and iteration, as well as to the writing, editing, and overall management of the project.

- **Yang Yue** (乐洋) led the project, first discovered the phenomenon where RL pass@k is surpassed by the base model, and proposed the idea; designed the experiments and partially conducted experiments; took primary responsibility for writing the manuscript.
- **Zhiqi Chen** conducted substantial experiments, including pass@k evaluation across models and benchmarks, and the perplexity analysis; contributed to discussions, figure creation, and manuscript review.
- **Rui Lu** contributed to inspiration of the idea and conceptualization of the project, story writing and manual check of AI reasoning trajectory.
- **Andrew Zhao** contributed to discussions on experimental design, proposed the perplexity-based analysis, and contributed to the early implementation of the RL training code.
- **Zhaokai Wang** contributed to discussions of RLVR's effect on reasoning boundary, writing, proofreading, and comprehensive manuscript review.
- **Yang Yue** (乐阳) contributed to the training of visual reasoning model, discussions, proofreading and figure refinement.
- **Gao Huang** & **Shiji Song** supervised the research, and assisted in writing the paper.

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

# Appendix

# Appendix Contents

# A Implementation Details

## A.1 RLVR Algorithms

To reduce memory and computational overhead, several critic-free variants have been proposed. GRPO [37] estimates the advantage with a normalized reward within a group of responses to the same question: $A_i = [r_i - \text{mean}(\mathbf{r})]/\text{std}(\mathbf{r})$, where $\mathbf{r} = \{r_1, \ldots, r_G\}$ denotes the set of rewards for a group of $G$ sampled responses. RLOO [39] instead adopts a leave-one-out baseline within each batch $\mathcal{B}$. Its advantage is defined as $A_i = r_i - \frac{1}{|\mathcal{B}|-1} \sum_{j \neq i} r_j$.

## A.2 Low-Variance *pass@k* Estimation

Directly computing pass@$k$ using only $k$ sampled outputs per problem can lead to high variance. To mitigate this, we follow the unbiased estimation method proposed by Chen *et al.* [14]. Specifically, for each problem $x_i$ from the evaluation dataset $\mathcal{D}$, we generate $n$ samples ($n \geq k$) and count the number of correct samples as $c_i$. The unbiased estimator of pass@$k$ over the dataset is given by:

$$\text{pass@}k := \mathbb{E}_{x_i \sim \mathcal{D}} \left[ 1 - \frac{\binom{n-c_i}{k}}{\binom{n}{k}} \right] \tag{2}$$

With this formulation, we can easily estimate pass@$k$ with low variance across all $k \leq n$.

In our experiments, we set $n$ to the largest (*i.e.*, rightmost) $k$ value in the pass@$k$ curves, typically 128, 256, or 1024. For example, in Figure 2, we use $n = 128$ for MATH500, Minerva, and GSM8K, and $n = 1024$ for AMC23 and AIME24. For the Olympiad benchmark, we set $n = 128$ for the Qwen models and $n = 1024$ for LLaMA-3.1-8B, due to its relatively lower base model capacity.

# B More Related Works

**Reinforcement Learning for LLM Reasoning.** Since the emergence of LLMs, the post-training phase has proven crucial to enhance problem solving and reasoning abilities [49]. This stage typically falls into three main categories: supervised fine-tuning using human-curated or distilled data [50], self-improvement iteration [51, 52], and reinforcement learning [49]. Previously, a reward model or preferences between responses were employed for reward modeling [49, 53]. Recently, Reinforcement Learning with Verifiable Rewards (RLVR) has gained significant traction as a method to improve the reasoning capabilities of LLMs in domains such as mathematics and programming [6, 37]. An encouraging landmark work is OpenAI's o1 model [1], which was among the first large-scale applications of RL for reasoning, achieving state-of-the-art results at the time of its release. Following this, Deepseek-R1 [2] became the first open-weight model to match or surpass the performance of o1. A significant innovation introduced with R1 is the "zero" setting, where reinforcement learning is applied directly to the base LLM, bypassing any intermediate supervised tuning. This approach inspired a wave of open-source efforts to replicate or extend R1's methodology and improve RL algorithms [18, 22–24]. In parallel, reinforcement learning has also gained attention in the multimodal domain, driving advancements in multimodal reasoning [29–31].

**Analysis of RLVR**. Although there are many excellent open-source works and algorithmic designs in the field of RLVR, there remains a lack of deep understanding regarding the root effects of RLVR on LLM reasoning abilities and its limitations when starting from the base model. Several studies [45–47] highlight that the reflective behaviors observed in R1-like models actually emerge from the base models, rather than being introduced by RLVR training. Dang *et al.* [48] observed a phenomenon similar to our findings: Pass@k deteriorates rapidly and fails to recover with reinforcement learning, but this was seen only in a limited experimental setup with Qwen-2.5-0.5B on GSM8K. More importantly, they did not explore the relationship between the base model and the RL model. In contrast, our paper conducts systematic and rigorous experiments to show that not only reflective behaviors but all reasoning paths are already embedded in the base model. We further demonstrate that RLVR does not elicit any new reasoning abilities beyond the base model.

# C  Discussion

In Section 3 and Section 4, we identified key limitations of RLVR in enhancing LLM reasoning capabilities. In this section, we explore possible underlying factors that may explain why RLVR remains bounded by the reasoning capacity of the base model.

**Discussion 1: Key Differences Between Traditional RL and RLVR for LLMs are Vast Action Space and Pretrained Priors.** Traditional RL such as AlphaGo Zero and the DQN series [8, 7, 54] can continuously improve the performance of a policy in environments like Go and Atari games *without an explicit upper bound*. There are two key differences between traditional RL and RLVR for LLMs. First, the action space in language models is exponentially larger than that of Go or Atari games [55]. RL algorithms were not originally designed to handle such a vast action space, which makes it nearly impossible to explore the reward signal effectively if training starts from scratch. Therefore, the second distinction is that RLVR for LLMs starts with a pretrained base model with useful prior, whereas traditional RL in Atari and GO games often begins from scratch. This pretrained prior guides the LLM in generating reasonable responses, making the exploration process significantly easier, and the policy can receive positive reward feedback.

**Discussion 2: Priors as a Double-Edged Sword in This Vast Action Space.** Since the sampling of responses is guided by the pretrained prior, the policy may struggle to explore new reasoning patterns beyond what the prior already provides. Specifically, in such a complex and highly combinatorial space, most responses generated during training are constrained by the base model's prior. Any sample deviating from the prior is highly likely to produce invalid or non-sensical outputs, leading to negative reward. As discussed in Section 2.1, policy gradient algorithms aim to maximize the log-likelihood of responses within the prior that receive positive rewards, while minimizing the likelihood of responses outside the prior that receive negative rewards. As a result, the trained policy tends to produce responses already present in the prior, constraining its reasoning ability within the boundaries of the base model. From this perspective, training RL models from a distilled model may temporarily provide a beneficial solution, as distillation helps inject a better prior.

**Possible Future Work.** Developing more efficient exploration strategies may be essential for navigating the vast action space, facilitating the discovery of out-of-prior reasoning patterns and the development of more capable reasoning models. Furthermore, current RLVR frameworks are limited to single-turn interactions with the verifier, lacking the ability to iteratively revise or improve based on environmental feedback. A multi-turn RL agent paradigm—with richer, ongoing interaction with a grounded environment—could enable models to generate novel experiences and learn from them. This agent paradigm has been described as the beginning of an "era of experience" [56].

# D  Detailed Experimental Results

## D.1  More Results on Mathematics and Coding

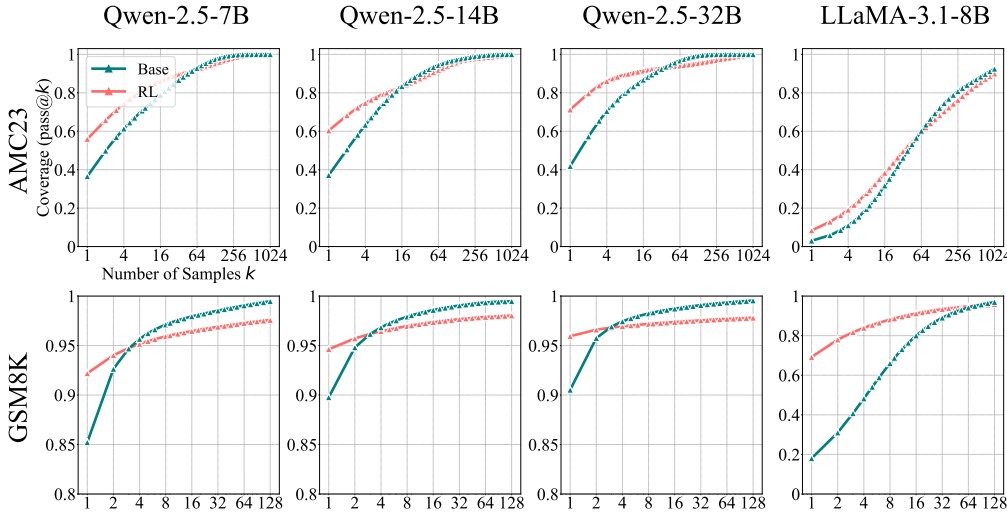

Figure 10: More results of SimpleRLZoo on GSM8K and AMC23.

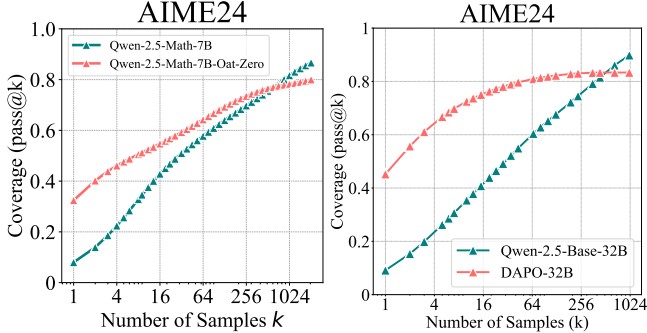

Figure 11: Oat-Zero-7B and DAPO-32B are evaluated on AIME24 and compared against their respective base models.

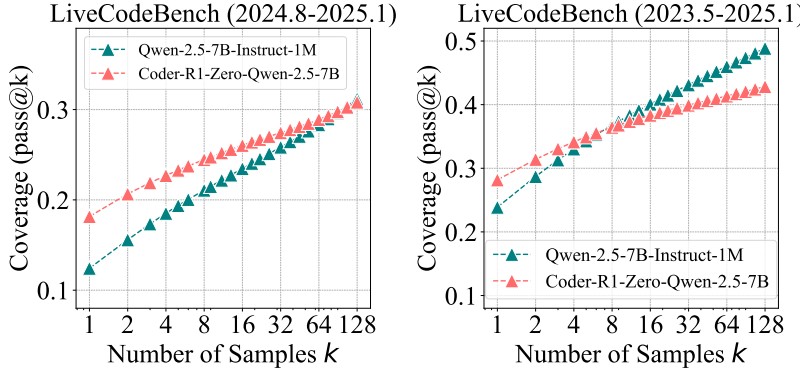

Figure 12: Coder-R1 on LiveCodeBench.

## D.2 Validity of Chain-of-Thought on AIME24

We manually check the CoTs for the most challenging AIME24 benchmark. To begin, we introduce a filtering mechanism designed to eliminate easily guessable problems. Specifically, we prompt Qwen2.5-7B-Base to answer questions directly, without using chain-of-thought reasoning, and sample answers multiple times. If a problem can be answered correctly with a low but non-zero probability (e.g., <5%), we consider it to be guessable and remove it. Problems that can be directly answered correctly with a high probability are retained, as they are likely easier and solvable using valid CoTs.

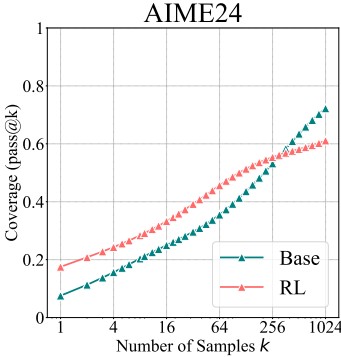

Figure 13: Pass@$k$ curves of the base and SimpleRLZoo-7B models in the filtered AIME24.

The base and RL model pass@$k$ curves on this filtered AIME24 can be found in Figure 13, showing a similar trending to previous results. Although this filtering method is heuristic, it proves to be effective. Applying it to AIME24 (30 questions) results in a subset of 18 questions. We then prompt the models to answer these filtered questions using CoT reasoning. Then we perform a manual inspection of all CoTs that led to correct answers on the hardest problems – those with an average accuracy below 5%. The base model answered 7 such questions, with 5 out of 6 containing *at least one* correct CoT (excluding one ambiguous case of correctness due to skipped reasoning steps). Similarly, the RL-trained model answered 6 questions, 4 of which included *at least one* correct CoT. These results suggest that even for the hardest questions in the challenging AIME24, base model can sample valid reasoning paths to solve the problems.

## D.3 Accuracy Distribution Visulization

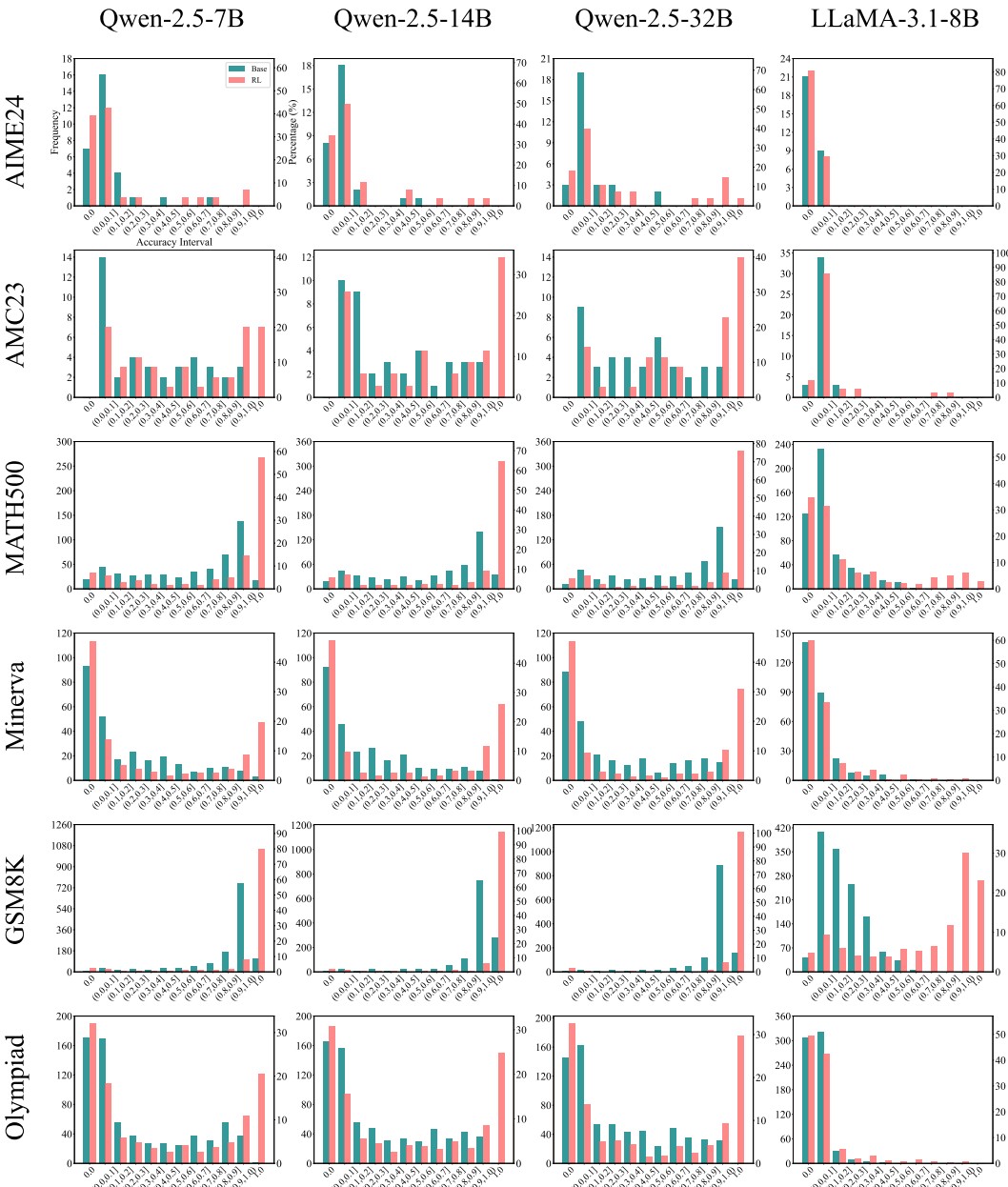

Figure 14: Accuracy histogram before and after RLVR with SimpleRLZoo models.

### D.4 Perplexity Analysis

To analyze how perplexity evolves over the course of RLVR training, we evaluated three RLVR checkpoints–early, middle, and final in Section 4.3 RL training. For each checkpoint, we sampled 32 responses per problem, computed the median among 32 perplexity values, and reported the average over the first 10 problems in the table. As expected, we observed that $\text{PPL}_{\text{Base}}(\mathbf{Y}_{\text{RL}}|x)$ gradually decreases as RL training progresses, indicating that RLVR mainly sharpens the distribution within the base model's prior rather than expanding beyond it.

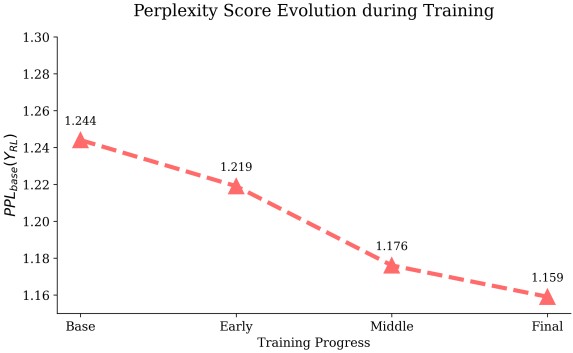

Figure 15: Perplexity Evolution during RL Training.

### D.5 Different RLVR Algorithms

We report several additional observations on different RLVR algorithms in Figure 8. First, DAPO achieves slightly higher pass@1 scores across all three datasets; however, its dynamic sampling strategy requires approximately $3 \sim 6\times$ more samples per batch during training compared to other algorithms. Moreover, its performance drops significantly at $k = 256$. Second, RLOO and Reinforce++ perform consistently well across the entire $k$ range (from 1 to 256), while maintaining efficient training costs, achieving a good balance between effectiveness and efficiency. Third, ReMax shows lower performance at both pass@1 and pass@256. We hypothesize that this is due to its use of the greedy response reward as the advantage baseline, which in the RLVR setting is binary (0 or 1) and highly variable. This likely results in unstable gradient updates during training.

Table 3: Detailed values for each point at pass@1 and pass@256 across different RL algorithms in Figure 8.

| Model | Omni-MATH-Train | | Omni-MATH-Test | | MATH500 | |
|---|---|---|---|---|---|---|
| | pass@1 | pass@256 | pass@1 | pass@256 | pass@1 | pass@256 |
| Qwen2.5-7B | 9.9 | 67.2 | 10.2 | 69.1 | 34.5 | 96.2 |
| GRPO | 26.1 | 66.3 | 25.1 | 68.3 | 74.4 | 97.2 |
| PPO | 27.2 | 65.8 | 26.8 | 69.2 | 75.2 | 97.2 |
| ReMax | 24.4 | 65.5 | 23.8 | 67.5 | 73.5 | 96.6 |
| RLOO | 28.6 | 66.4 | **28.1** | 69.2 | 75.0 | **97.4** |
| Reinforce++ | 28.2 | **67.7** | **28.0** | **69.7** | 75.4 | 96.8 |
| DAPO | **31.4** | 66.1 | 26.5 | 67.0 | **75.6** | 96.4 |

Table 4: Detailed values at pass@1 and pass@256 across different RL training steps in Figure 1 (right).

| Model | Omni-MATH-Train | | Omni-MATH-Test | | MATH500 | |
|---|---|---|---|---|---|---|
| | pass@1 | pass@256 | pass@1 | pass@256 | pass@1 | pass@256 |
| Qwen2.5-7B | 9.9 | **67.2** | 10.2 | **69.1** | 34.5 | 96.2 |
| GRPO-step150 | 26.1 | 66.3 | 25.1 | 68.3 | 74.4 | **97.2** |
| GRPO-step300 | 33.6 | 65.3 | 27.1 | 66.6 | 75.4 | 96.0 |
| GRPO-step450 | **42.5** | 64.3 | **28.3** | 63.9 | **76.3** | 95.4 |

## D.6 Effects of KL and Rollout Number

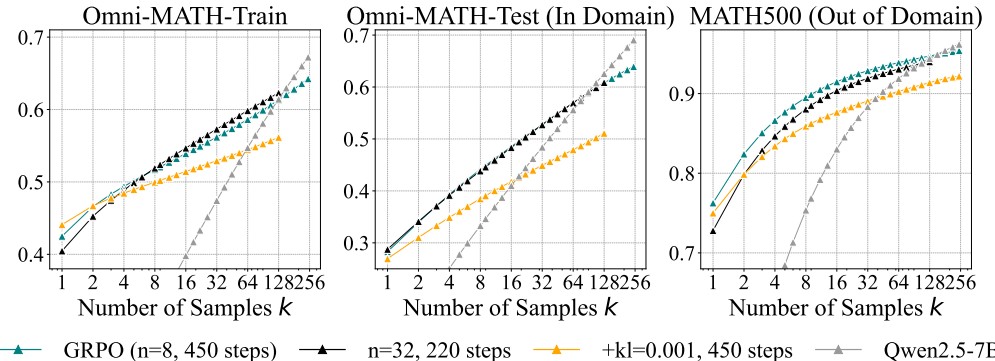

Figure 16: **Ablation Study on KL Loss and Rollout Number** $n$**.** For increasing $n$ from 8 to 32, we keep the prompt batch size unchanged, which results in increased computation per training step. Due to resource constraints, we train for only 220 steps under this setting, leading to lower pass@1 as the model has not yet converged. Nevertheless, the model with $n = 32$ achieves a higher pass@128, highlighting the positive effect of larger rollout numbers in improving pass@$k$ at higher values of $k$.

## D.7 Solvable Problem Coverage Analysis

Table 2 reports the fraction of problems categorized as four conditions: (1) both models solve the problem at least once, (2) only the base model solves it, (3) only the RLVR model solves it, and (4) neither model solves it in any of the $k$ samples. It highlights that there are many cases where the base model solves a problem but RLVR fails (type 2), and very few where RLVR succeeds while the base does not (type 3). Even in the rare type 3 cases (e.g., 1% or about 5 problems in MATH500), the base model is able to solve all of them when sampling 1024 times. These results support our conclusion that RLVR rarely solves problems the base model cannot and generally results in reduced coverage.

Table 5: Indices of solvable problems in AIME24 (starting from 0). An approximate subset relationship can be observed: most problems solved by the RL model are also solvable by the base model.

| Models | Problem Indices |
|---|---|
| Qwen2.5-7B-Base | 0, 1, 4, 6, 7, 8, 9, 11, 12, 14, 15, 16, 17, 18, 19, 22, 23, 24, 25, 26, 27, 28, 29 |
| SimpleRL-Qwen2.5-7B | 0, 1, 6, 7, 8, 9, 12, 14, 15, 16, 18, 22, 23, 24, 25, 26, 27, 28, 29 |

Table 6: Indices of solvable problems in LiveCodeBench (ranging from 400 to 450, starting from 0).

| Model | Solvable Problem Indices |
|---|---|
| Qwen2.5-7B-Instruct-1M | 400, 402, 403, 407, 409, 412, 413, 417, 418, 419, 422, 423, 427, 432, 433, 436, 438, 439, 440, 444, 445, 448, 449 |
| Coder-R1 | 400, 402, 403, 407, 412, 413, 417, 418, 419, 422, 423, 427, 430, 433, 438, 439, 440, 444, 445, 449 |

## D.8 Temperature and Entropy Analysis

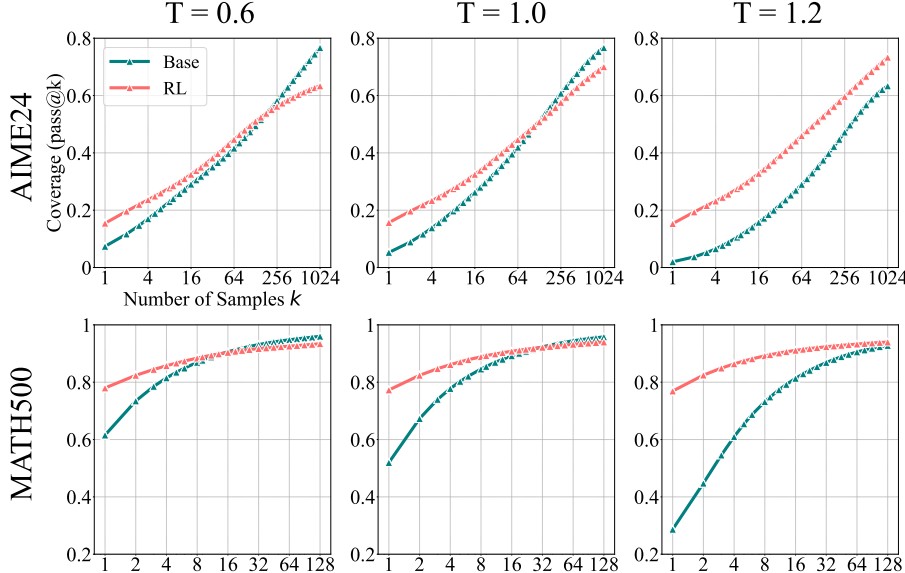

Figure 17: We found that the base model's performance drops when the temperature exceeds 1.0, as it tends to generate more random and less coherent tokens. In contrast, the RL model's performance remains relatively stable across different temperature settings. Therefore, we use $T = 0.6$ in the main experiments, as it allows both models to demonstrate their best reasoning performance.

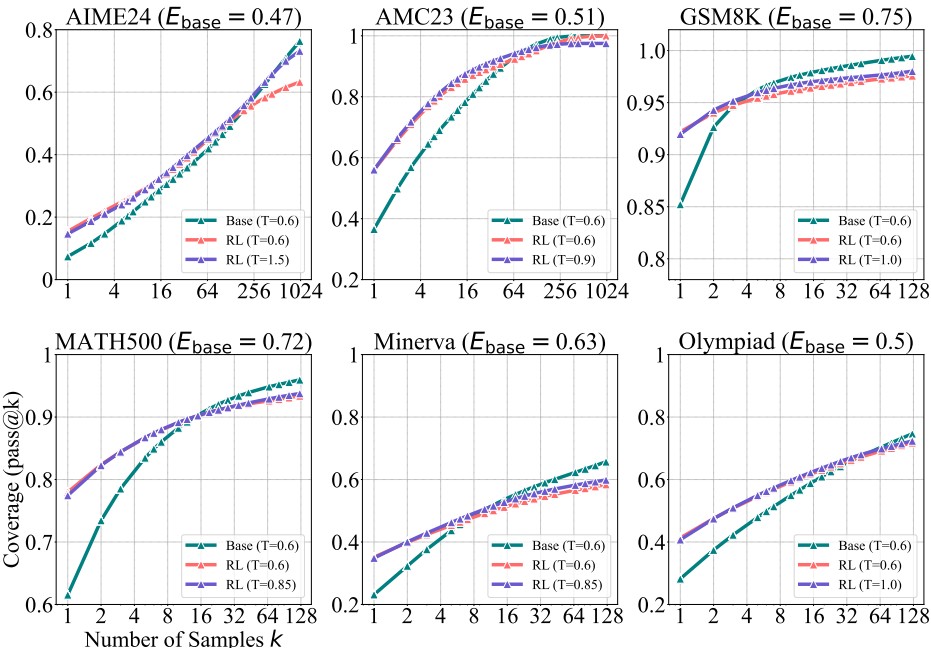

Figure 18: **Comparison of Base and RLVR Models with Matched Output Entropy.** We evaluate the base model (Qwen2.5-7B) on each dataset using temperature $T = 0.6$ and report its output entropy $E_{\text{base}}$ in the title of each figure. To enable a fair comparison, we increase the temperature of the RLVR model (SimpleRLZoo) until its output entropy approximately matches $E_{\text{base}}$. For example, on AMC23, we set $T = 0.9$ to achieve $E_{\text{RL}} = 0.47$. We also include RLVR results at $T = 0.6$ as an additional baseline, which has lower entropy—e.g., 0.22 on AMC23 and 0.33 on MATH500.

## D.9 Training Dynamics

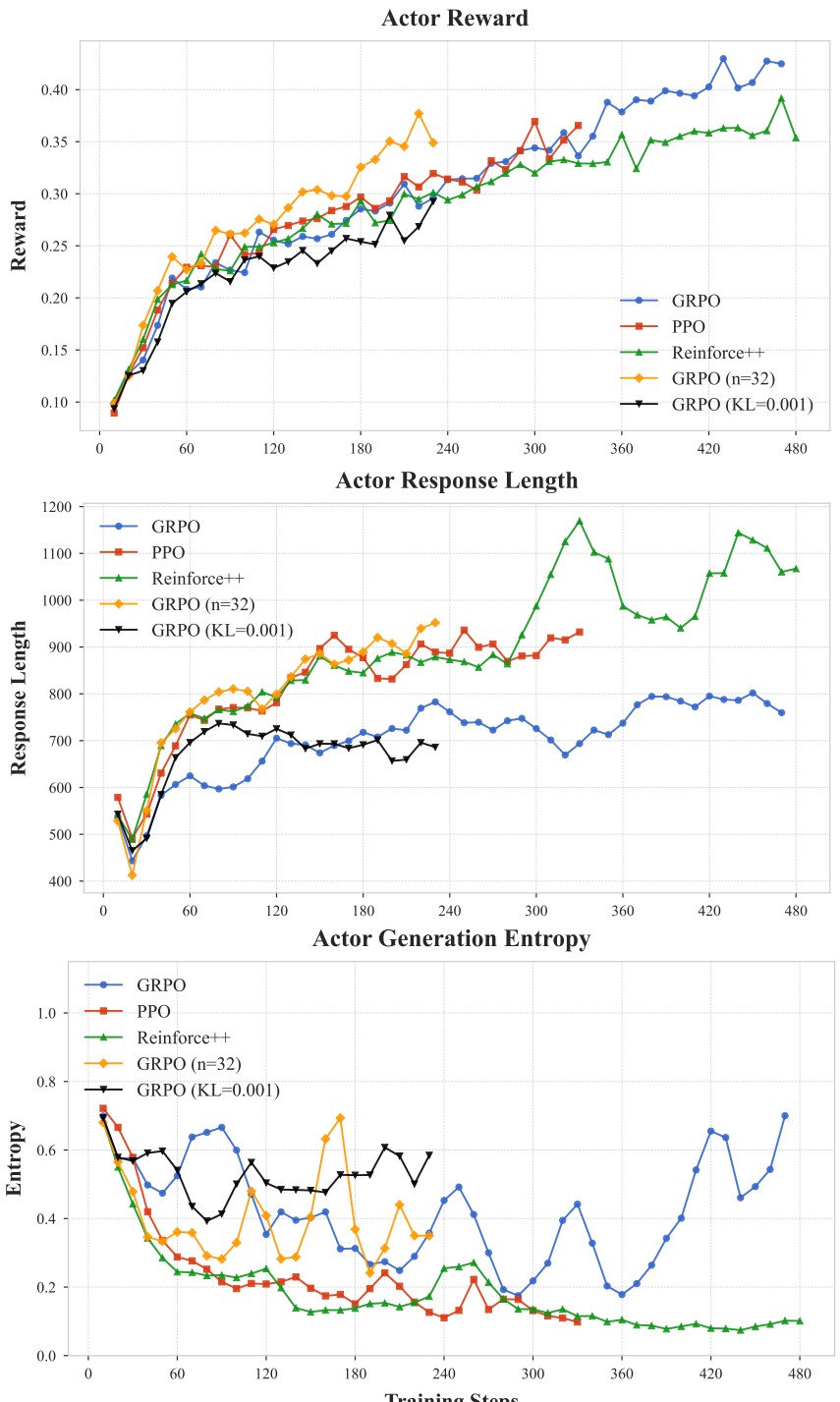

Figure 19: The curves of training reward, response length, and generation entropy during training, corresponding to experiments in Section 4.

## D.10 CoT Case Analysis

**(AIME24 Question No.16)**
Among the 900 residents of Aimeville, there are 195 who own a diamond ring, 367 who own a set of golf clubs, and 562 who own a garden spade. In addition, each of the 900 residents owns a bag of candy hearts. There are 437 residents who own exactly two of these things, and 234 residents who own exactly three of these things. Find the number of residents of Aimeville who own all four of these things.

**Qwen-2.5-7B Answer**                                    Response Length: 1058

To solve this problem, we can use the principle of inclusion-exclusion. We will denote the number of residents who own diamond rings, golf clubs, garden spades, and candy hearts as D, G, S, and C, respectively. We are given the following information:
- D = 195
- G = 367
- S = 562
- C = 900 (since every resident owns a bag of candy hearts)
We are also given that 437 residents own exactly two of these things and 234 residents own exactly three of these things.

Let's denote the number of residents who own all four of these things as X. We need to find X.

First, we need to find the number of residents who own at least one of these things. This can be calculated using the principle of inclusion-exclusion:

$$\text{Total} = D + G + S + C - (\text{number of residents who own exactly two things})$$
$$-2 \times (\text{number of residents who own exactly three things}) - 3 \times (\text{number of residents who own all four things})$$

Since there are 900 residents in total, we can write the equation as:

$$900 = 195 + 367 + 562 + 900 - 437 - 2 \times 234 - 3 \times X$$

Now we need to solve for X:

$$900 = 195 + 367 + 562 + 900 - 437 - 468 - 3 \times X$$
$$900 = 1717 - 905 - 3 \times X$$
$$900 = 812 - 3 \times X$$
$$3 \times X = 812 - 900$$
$$3 \times X = -88$$
$$X = -\frac{88}{3}$$

**Since the number of residents cannot be negative, there is an error in our calculations. Let's re-evaluate the problem.**

We know that there are 437 residents who own exactly two things and 234 residents who own exactly three things. We can subtract these from the total number of residents who own at least one thing to find the number of residents who own exactly one thing:

$$\text{Number of residents who own exactly one thing} = (D + G + S + C) - (\text{number of residents who own exactly two things})$$
$$-2 \times (\text{number of residents who own exactly three things}) - 3 \times (\text{number of residents who own all four things})$$

Let's denote the number of residents who own exactly one thing as Y:

$$Y = (195 + 367 + 562 + 900) - 437 - 2 \times 234 - 3 \times X$$
$$Y = 1924 - 905 - 3 \times X$$
$$Y = 1019 - 3 \times X$$

Since every resident owns at least one thing, we can write the equation:
$$900 = Y + 437 + 2 \times 234 + 3 \times X$$

Now we need to solve for X:

$$900 = (1019 - 3 \times X) + 437 + 468 + 3 \times X$$
$$900 = 1019 + 437 + 468$$
$$900 = 1924$$

**This equation is not true, which means there is still an error in our calculations. Let's try a different approach.**

We can use the fact that the total number of residents who own at least one thing is 900. We can write the equation:

$$900 = (\text{number of residents who own exactly one thing}) + (\text{number of residents who own exactly two things})$$
$$+(\text{number of residents who own exactly three things}) + (\text{number of residents who own all four things})$$

Let's denote the number of residents who own exactly one thing as Y:

$$900 = Y + 437 + 2 \times 234 + X$$
$$900 = Y + 805 + X$$

Now we need to find Y. We can use the fact that the total number of residents who own at least one thing is equal to the sum of the residents who own each item minus the residents who own at least two items:

$$900 = (195 + 367 + 562 + 900) - (\text{number of residents who own exactly two things})$$
$$-2 \times (\text{number of residents who own exactly three things}) - 3 \times (\text{number of residents who own all four things})$$
$$900 = 1924 - 437 - 2 \times 234 - 3 \times X$$
$$900 = 1924 - 805 - 3 \times X$$
$$900 = 1119 - 3 \times X$$

Now we can solve for X:

$$3 \times X = 1119 - 900$$
$$3 \times X = 219$$
$$X = 73$$

So, the number of residents of Aimeville who own all four of these things is $\boxed{73}$.

Figure 20: Qwen2.5-Base-7B Correct Response - Case 1.

**(AIME24 Question No.24)**

Find the number of ways to place a digit in each cell of a 2x3 grid so that the sum of the two numbers formed by reading left to right is 999, and the sum of the three numbers formed by reading top to bottom is 99. The grid below is an example of such an arrangement because $8 + 991 = 999$ and $9 + 9 + 81 = 99$.

| 0 | 0 | 8 |
|---|---|---|
| 9 | 9 | 1 |

## Qwen-2.5-7B Answer                                    Response Length: 967

To solve this problem, we need to determine the number of ways to place digits in a 2x3 grid such that the sum of the two numbers formed by reading left to right is 999, and the sum of the three numbers formed by reading top to bottom is 99. Let's break this down step by step.

1. **Define the grid:**
   Let the grid be represented as:
   $$a \quad b \quad c$$
   $$d \quad e \quad f$$
   The numbers formed by reading left to right are $100a + 10b + c$ and $100d + 10e + f$. The numbers formed by reading top to bottom are $10a + d$, $10b + e$, and $10c + f$.

2. **Set up the equations:**
   - The sum of the two numbers formed by reading left to right is 999:
   $$100a + 10b + c + 100d + 10e + f = 999$$
   - The sum of the three numbers formed by reading top to bottom is 99:
   $$10a + d + 10b + e + 10c + f = 99$$

3. **Simplify the equations:**
   - From the first equation:
   $$100a + 10b + c + 100d + 10e + f = 999$$
   This implies:
   $$a + d = 9, b + e = 9, c + f = 9$$
   - From the second equation:
   $$10a + d + 10b + e + 10c + f = 99$$
   This is consistent with the previous simplification since:
   $$10(a + b + c) + (d + e + f) = 99$$
   Given $a + d = 9$, $b + e = 9$, and $c + f = 9$, it follows that:
   $$10(a + b + c) + (9 - a) + (9 - b) + (9 - c) = 99$$
   Simplifying, we get:
   $$10(a + b + c) + 27 - (a + b + c) = 99$$
   $$9(a + b + c) + 27 = 99$$
   $$9(a + b + c) = 72$$
   $$a + b + c = 8$$
   Since $a + d = 9$, $b + e = 9$, and $c + f = 9$, and $a + b + c = 8$, it follows that:
   $$d = 9 - a, e = 9 - b, f = 9 - c$$

4. **Determine the possible values:**
   - $a, b, c$ must be digits (0-9) such that $a + b + c = 8$.
   - For each valid combination of $a, b, c, d, e, f$ are uniquely determined as $d = 9 - a, e = 9 - b, f = 9 - c$.

5. **Count the combinations:**
   - We need to count the number of non-negative integer solutions to $a + b + c = 8$ with $0 \leq a, b, c \leq 9$.
   - This is a classic "stars and bars" problem with an additional constraint. The number of solutions is given by the binomial coefficient:
   $$\binom{8 + 3 - 1}{3 - 1} = \binom{10}{2} = 45$$

Therefore, the number of ways to place the digits in the grid is $\boxed{45}$.

Figure 21: Qwen2.5-Base-7B Correct Response - Case 2.

# E    Prompt Templates

We provide the prompt templates used for training and evaluation in our experiments. The prompt for SimpleRL training and evaluation is shown in Figure 22, while the prompt for Oat-Zero is shown in Figure 23. For Code-R1 training, prompt in Figure 24 is adopted. For Code-R1 evaluation, we follow the original codebase and adopt the default templates from the benchmarks, including LiveCodeBench prompt (Figure 25), HumanEval+, and MBPP+ prompt (Figure 26). The prompt used for EasyR1 training and evaluation is shown in Figure 27. For VeRL-trained RL models, as discussed in Section 4.3 and Section 4.4, the training and evaluation prompts are provided in Figure 28. For evaluating Mistral and Magistral models on AIME24/25, prompts are provided in Figure 29. To ensure a fair comparison, the base models use the same prompts as their corresponding RL-trained counterparts during evaluation.

---

**SimpleRL Prompt**

<|im_start|>system
You are a helpful assistant.<|im_end|>
<|im_start|>user
{question}
Please reason step by step, and put your final answer within\\boxed{{}}.<|im_end|>
<|im_start|>assistant

Figure 22: Prompt for SimpleRL Training and Evaluation. The base model uses the same prompt as the RL model during evaluation.

---

**Oat Prompt**

<|im_start|>system
Please reason step by step, and put your final answer within \\boxed{}.<|im_end|>
<|im_start|>user
{question}<|im_end|>
<|im_start|>assistant

Figure 23: Prompt for Oat-Zero training and evaluation.

---

**Code-R1 Prompt**

<|im_start|>system
You are a helpful programming assistant. The user will ask you a question and you as the assistant solve it. The assistant first thinks how to solve the task through reasoning and then provides the user with the final answer. The reasoning process and answer are enclosed within <think>...</think> and <answer>...</answer> tags, respectively.<|im_end|>
<|im_start|>user
{question}<|im_end|>
<|im_start|>assistant

Figure 24: Prompt for Code-R1 training.

**LiveCodeBench (Code Generation) Prompt**

```
<|im_start|>system
You are a helpful assistant.<|im_end|>
<|im_start|>user
You will be given a question (problem specification) and will generate a correct
Python program that matches the specification and passes all tests. You will NOT
return anything except for the program.

Question: {question.question_content}

{ if question.starter_code }
### Format:
{PromptConstants.FORMATTING_MESSAGE_WITH_STARTER_CODE}
```python
{question.starter_code}
```

<|im_end|>
{ else }
### Format:
{PromptConstants.FORMATTING_MESSAGE_WITHOUT_STARTER_CODE}
```python
# YOUR CODE HERE
```

<|im_end|>
<|im_start|>assistant
```python
```

Figure 25: Since Code-R1 does not specify an evaluation prompt, we adopt the original Live-CodeBench evaluation prompt. To encourage both the base and RL-trained models to generate code, we append ```python to the end of the prompt. Using this setup, we reproduce a pass@1 score of 28.6, which is close to the reported 29.7.

**HumanEval+ & MBPP+ Prompt**

```
<|im_start|>system
You are a helpful assistant.<|im_end|>
<|im_start|>user
Please provide a self-contained Python script that solves the following problem in a
markdown code block:
```
{python_task_prompt}
```

<|im_end|>
<|im_start|>assistant
Below is a Python script with a self-contained function that solves the problem and
passes corresponding tests:
```python
```

Figure 26: Prompt for Code-R1 Evaluation on HumanEval+ and MBPP+.

Figure 27: Prompt for EasyR1 training and evaluation.

Figure 28: Prompt for VeRL training on Omni-math-train and evaluation on Omni-math-eval and MATH500.

Figure 29: Prompt for Mistral and Magistral model evaluation.

# F   Broader Impacts

The potential negative social impacts of our method align with those typically associated with general LLM reasoning technologies. We emphasize the importance of adhering to the principles of fair and safe deployment in LLM systems.

