# OpenReview forum: "Does Reinforcement Learning Really Incentivize Reasoning Capacity in LLMs Beyond the Base Model?"
_NeurIPS.cc/2025/Conference — NeurIPS 2025 oral_

### Official Review · Reviewer_KdU6 · 2025-06-17

**Clarity:** 4
**Significance:** 4
**Originality:** 3
**Rating:** 6
**Confidence:** 3

**Summary:**

This paper thoroughly investigates the effect of RLVR on the reasoning ability of LLMs, measured by a pass@k metric under different sampling budget k.
They find that RLVR actually does not improve the reasoning performance upper bound of LLMs, but instead make the LLM sample promising CoTs with higher probability,at the cost of a more restricted sampling distribution and correspondingly lower performance than the baseline LLMs that sample from a wider distribution when given a larger sampling budget.
These findings provide a critical reflection on people's common belief that RLVR can improve the reasoning ability of LLMs, which are very straightforward to understand and insightful.

**Questions:**

1. The models evaluated in this paper mainly have ~7B parameters, with a maximal evaluated model size of 32B. I think an important question here is if the conclusion drawn from the paper still holds when we scale up the model size? Some evaluation results on larger models like o1 and DeepSeek-R1-Zero will be quite helpful here. I understand that this may introduce significantly higher computational cost, so some priliminary results would be great enough, otherwise this should be clearly stated as a limitation of this paper due to computational constraints and call for future work.
Moreover, this paper gives a great discussion on the limitations of existing RLVR methods and the reasons, but a discussion on the limitations of this paper itself seems to be missed.
2. Another dimension of variability that I'm interested about is how the gap in the pass@k curves changes w.r.t. the difficulty of the evaluation tasks. Intuitively, I think more challenging task will reduce the gap between base model and RLVR model when k is large, as it's harder to sample the correct CoT from a prior distribution. I'm an RL instead of LLM researcher, so I'm not sure if there are some clear order of difficulty in the evaluation tasks used in the paper. If yes, I think some analysis on how task difficulty influences the results here would make this paper even more valuable to the community.
3. If I understand correctly, Most pass@k curves reported in the paper are the results on some hold-out test sets that are not used for RLVR training right? If so, how would the pass@k curves look like on the training tasks?
4. The Perplexity analysis in Figure 6 is very interesting. But it only shows the perplexity score at the end of RLVR training. I would suggest to further add a figure to show how the perplexity score evolves during the RLVR training process to better support your analysis. E.g., based on your analysis, I would expect that $PPL_{RL}(Y_{base})$ to start from a uniform distribution and gradually converge to two clusters after training, while $PPL_{base}(Y_{RL})$ gradually have lower scores during training. Furthermore,  the very high score of $PPL_{base}(Y_{GT})$ indicates that a larger LLM like o1 may behave very differently from smaller ones, which again motivate my first point in this section.

Overall, I'm happy to further raise my rating if the authors can help answer or address these questions, especially the first point on scaling with model size.

**Ethical Concerns:**

["NO or VERY MINOR ethics concerns only"]

**Final Justification:**

The authors have addressed most of my concerns during rebuttal with supportive experiments and analysis. So I'm happy to further raise my score.

**Limitations:**

I don't find a clear discussion of the limitations of this paper in the text, and would suggest the authors to add it, e.g., the first point I mention in the question section.

**Quality:**

4

**Strengths And Weaknesses:**

Strengths:
1. Clear and insightful motivation, which leads to conclusions that could be very beneficial for the RLVR community;
2. The experiments are designed in a very logical way that start from some surprising benchmark results questioning some common belief before, then give detailed analysis on the potential reasons of the issue with further experiments that well support these hypothesis.
3. The paper is well written and structured, with clear definitions, thorough experimental evaluation and details.

Weakness:
See my questions in the next section.

---

> ### Author Rebuttal · Authors · 2025-07-31
>
> # Response
>
> Thanks for supporting the acceptance of our work. Please find our responses to your questions below.
>
> ### `Q1: Does the conclusion drawn from the paper still hold when we scale up the model size? some priliminary results would be great enough`
>
> Thank you for the insightful question. We also care deeply about whether scaling up model size affects our conclusions.
> **After careful investigation, we found that only three models over 70B parameters have both base and RLVR-trained versions publicly available**: DeepSeek-R1-Zero, Tulu-3-70B [1], and Magistral [2]. We summarize them in the table below.
> For many other large models, isolating the effect of RLVR is not feasible. For example:
>
> - **GPT-o1**: The base model is not accessible, and training details are unclear; the final model may not rely solely on RLVR.
> - **Qwen3-235B**: The reasoning model undergoes multiple stages, including RLVR and long-CoT SFT, making it impossible to isolate the effect of RLVR alone.
> - **LLaMA-3-70B**: no corresponding RLVR version exists.
>
> | Model Pair   | Size    | API Access | Weights Access |
> | - | ---| -- | -- |
> | DeepSeek-V3-Base / R1-Zero | 671B    | ✗          | ✓                 |
> | Tulu-3-SFT / Tulu-3-RLVR  | 70B     | ✗          | ✓                 |
> | Mistral-Medium-3 / Magistral-Medium       |    undisclosed*    | ✓          | ✗                 |
>
> *: **undisclosed size but reasoning performance close to DeepSeek-R1-671B**
>
> **For Deepseek-R1-Zero**, we have done our best to make it run with limited resources (two 8xH100 GPU nodes). However, without expert engineering optimization, the throughput remains very low, around 50 tokens/s at a maximum sequence length of 32k. Runnings on AIME24 with 1024 samplings would take approximately 4 months, which makes it currently infeasible. We are actively seeking additional resources and hope to resume the process later.
>
> Thus, we turned to Tulu-3-70B and Magistral.
>
> **For Tulu-3-70B**, which we hosted locally with 8 H100 GPUs, we conducted evaluations on the first 100 subsampled problems from the MATH500 benchmark due to high computational cost. The results below are consistent with our main conclusion.
>
> ### **Tulu3 on Math500**
>
> | Model          | k=1  | k=4  | k=16 | k=32 |
> | -------------- | ---- | ---- | ---- | ---- |
> | Tulu-3-SFT     | 58.6% | 77.2% | 88.1% | 91.0% |
> | Tulu-3-RLVR    | 64.4% | 80.9% | 87.7% | 89.0% |
> | Δ (RLVR - SFT) | +5.8% | +3.7% | -0.4% | -2.0% |
>
> **For Magistral, one of the most advanced reasoning models whose. reasoning performance is colse to Deepseek-R1**, we queried the API using a maximum context length of 40k. Again, we observed RLVR's great improvements at low k, but little to no gain at larger k. Specifically, at k = 1, the RLVR model solves approximately 7 more problems than the base on AIME24 and 8 more on AIME25. However, as k grows, the performance gap steadily narrows. **The trend observed in this preliminary experiment further supports that our conclusion holds even for a frontier-level reasoning model.**
>
> ### **Magistral on AIME24**
> | *k*  | 1      | 4      | 64     | 256   | 1024  |
> | - | ------ | - | - | - | - |
> | Mistral-Medium-3 (Base) | 53.5%  | 69.7%  | 80.0%  | 87.6% | 90.0% |
> | Magistral-Medium (RLVR)| 76.6%  | 88.8%  | 90.0%  | 93.3% | 93.3% |
> | Δ (RLVR - Base) | +23.1% | +19.1% | +10.0% | +5.7% | +3.3% |
>
>
> ### **Magistral on AIME25**
> | *k*  | 1      | 4      | 64     | 256    | 1024  |
> | - | - | - | - | - | -|
> | Mistral-Medium-3 (Base) | 33.3%  | 51.5%  | 73.2%  | 81.3%  | 93.3% |
> | Magistral-Medium (RLVR) | 62.0%  | 75.8%  | 88.7%  | 92.1%  | 93.3% |
> | Δ (RLVR - Base)         | +26.7% | +24.3% | +15.5% | +10.8% | 0.0% |
>
> We will include these results  in the revision and explicitly discuss the scalability limitation. We also appreciate your suggestions, which help us strengthen the paper. A more comprehensive evaluation on very large models is currently constrained by compute resources, but this is an important direction for future work.
>
> [1] Allen Institute for AI. "Tulu 3: Pushing frontiers in open language model post-training."
>
> [2] Mistral AI. "Magistral." arXiv:2506.10910.
>
> ###  `Q2:  how the gap in the pass@k curves changes w.r.t. the difficulty of the evaluation tasks.`
>
> Thank you for the thoughtful question. To explore this, we analyzed the MATH500 benchmark, which includes difficulty annotations (Levels 1–5) based on AoPS (Art of Problem Solving) criteria. These levels are based on  knowledge domain, solution technique, and complexity. Math500 dataset contains 43, 90, 105, 128, and 134 problems at Levels 1 to 5, respectively. Results are as follows:
>
> ### **level 1**
>
> | pass@k|1 | 4 | 16 | 64 | 128 |
> | --- | --- | --- | --- | --- | --- |
> |Base| 86.4% | 96.2% | 97.9% | 99.4% | 100.0% |
> |RLVR| 93.4% | 94.7% | 95.3% | 95.3% | 95.3% |
> |Δ(Base-RLVR)| -7.0% | 1.5% | 2.6% | 4.1% | 4.7% |
>
> ### **level 2**
>
> | pass@k|1 | 4 | 16 | 64 | 128 |
> | --- | --- | --- | --- | --- | --- |
> |Base| 79.4% | 93.3% | 97.2% | 98.3% | 98.9% |
> |RLVR| 91.1% | 94.7% | 96.8% | 97.8% | 97.8% |
> |Δ(Base-RLVR)| -11.7% | -1.4% | 0.4% | 0.5% | 1.1% |
>
> ### **level 3**
>
> | pass@k|1 | 4 | 16 | 64 | 128 |
> | - | - | - | - | - | - |
> |Base| 74.9% | 93.2% | 98.1% | 99.9% | 100.0% |
> |RLVR| 89.4% | 96.0% | 98.6% | 99.0% | 99.0% |
> |Δ(Base-RLVR)| -14.5% | -2.8% | -0.5% | 0.9% | 1.0% |
>
> ### **level 4**
>
> | pass@k|1 | 4 | 16 | 64 | 128 |
> | - | - | - | - | - | - |
> |Base| 57.5% | 80.2% | 91.0% | 95.8% | 97.7% |
> |RLVR| 76.5% | 85.7% | 91.0% | 93.2% | 94.5% |
> |Δ(Base-RLVR)| -19.0% | -5.5% | 0.0% | 2.6% | 3.2% |
>
> ### **level 5**
>
> | pass@k|1 | 4 | 16 | 64 | 128 |
> | - | - | - | - | - | - |
> |Base| 34.9% | 60.8% | 77.7% | 86.0% | 88.1% |
> |RLVR| 56.6% | 68.8% | 77.4% | 82.5% | 84.3% |
> |Δ(Base-RLVR)| -21.7% | -8.0% | 0.3% | 3.5% | 3.8% |
>
> Preliminary observations:
>
> - **At k=1, the gap between base and RLVR increases with difficulty level**. Easy problems are often solved by the base on the first try. For harder problems, it's more difficult for the base to sample the correct CoT initially, while RLVR better optimizes sample efficiency. As a result, the crossover point—where the base overtakes RLVR—shifts with difficulty: around k=2 for Level 1, and k=16 for Levels 4–5.  These aligns well with your intuition that more difficult tasks require more samples to retrieve correct CoTs from the prior for base model.
> - However, at very large k, the gap becomes less correlated with difficulty level. The gap depends on how many problems the base model can eventually solve while the RLVR model cannot, due to reduced output diversity during RLVR training. How this relates to task difficulty is still unclear and warrants further investigation.
>
> We appreciate this suggestion and will include this analysis in the revision.
>
> ###  `Q3: Most pass@k curves reported in the paper are the results on some hold-out test sets that are not used for RLVR training right? If so, how would the pass@k curves look like on the training tasks?`
>
> Yes, the most pass@k curves are reported on hold-out test sets that were not used during RLVR training.
>
> In Sections 4.3 and 4.4, we present pass@k curves on both training and test sets as fig.14 shows, and we find the trends to be highly consistent to the test set.
>
> To further support this, we subsampled 100 examples from the training set for SimpleRLZoo-7B and plotted the pass@k curves. **The results on the training set mirror those on the test set**: RLVR outperforms the base model at small k, but is overtaken by the base model at larger k.
>
> ### **Training set**
>
> | k | 1 | 4 | 16 | 64 | 128|
> | - | - | -| -| - | - |
> | Base  | 43.4% | 58.4% | 70.0% | 80.2% | 84.5% |
> | RLVR | 52.7% | 57.6% | 61.4 | 64.9% | 66.7% |
> | Δ (Base - RLVR) | -9.3% | 0.8%  | 8.6%  | 15.3% | 17.8% |
>
>
> ### **Test set (Math500)**
>
> | k | 1 | 4 | 16 | 64 | 128|
> | -- | - | -| - | - | - |
> | Base | 61.5%  | 81.5% | 90.6% | 94.8% | 96.0% |
> | RLVR  | 78.0%  | 85.7% | 90.4% | 92.6% | 93.4% |
> | Δ (Base - RLVR) | -16.5% | -4.2% | 0.2%  | 2.2%  | 2.6%  |
>
>
> ### `Q4: further add a figure to show how the perplexity score evolves during the RLVR training process`
>
> Thank you for the thoughtful suggestion. To analyze how perplexity evolves over the course of RLVR training, we evaluated three RLVR checkpoints—early, middle, and final. For each checkpoint, we sampled 32 responses per problem, computed the median among 32 perplexity values, and reported the average over the first 10 problems in the table (since images are not allowed in rebuttals).
>
> **As expected, we observed that $PPL_{base}(Y_{RL})$ gradually decreases as RL training progresses**, indicating that RLVR mainly sharpens the distribution within the base model’s prior rather than expanding beyond it.
>
> | Metric | Value |
> | - | - |
> | $PPL_{base}(Y_{base})$   | 1.244 |
> | $PPL_{base}(Y_{early})$  | 1.219 |
> | $PPL_{base}(Y_{middle})$ | 1.176 |
> | $PPL_{base}(Y_{final})$  | 1.159 |
>
> **For $PPL_{RL}(Y_{base})$, we observed the emergence of two clusters over time**, consistent with your expectation. One cluster shows a slow decrease in perplexity, corresponding to base samples that remain within RLVR’s sharpened distribution. The other cluster exhibits a rapid increase in perplexity, reflecting base samples that fall outside the RLVR’s sharpened distribution.
>
> | Metric  | Cluster 1 | Cluster 2 |
> | - | --- | - |
> | \$PPL\_{RL-early}(Y\_{base})\$  | 1.195     | 1.347     |
> | \$PPL\_{RL-middle}(Y\_{base})\$ | 1.193     | 1.368     |
> | \$PPL\_{RL-final}(Y\_{base})\$  | 1.190     | 1.506     |
>
> Since this dynamic behavior is well aligned with the prediction from our conclusion, we believe this analysis further supports our claims.
>
> ### `conclusion`
>
> We hope our responses, especially regarding the first point on scaling to larger model sizes, have addressed your concerns and strengthened your confidence in the paper. If you have any further questions or suggestions, please don’t hesitate to let us know.

---

> ### Comment · Reviewer_KdU6 · 2025-08-01
>
> Dear authors,
>
> Thanks for your thorough rebuttal! The new results and analysis address most of my concerns. So I'm happy to further raise my score to 6.
>
> One further question that I have is that could the authors share some thoughts on this paper: [Reinforcement Learning with Verifiable Rewards Implicitly Incentivizes Correct Reasoning in Base LLMs](https://arxiv.org/abs/2506.14245), which draws an opposite conclusion as yours by using a more fine-grained evaluation metric called  CoT-Pass@k. Your reply to this question will not change my score on the submission, as this new paper is online after NeurIPS submission deadline. But I think it would be a very helpful discussion to the community to discuss about these two papers that drawer opposite conclusions.

---

> ### Author Response · Authors · 2025-08-03
>
> Thank you sincerely for your timely follow-up and continued support of our work.
> We have carefully read the paper you mentioned. We believe the difference may stem from several key factors:
>
> 1. Verifier Limitations:
> The CoT-Pass@k metric in that paper relies on **an 8B verifier model** (DeepSeek-R1-0528-Qwen3-8B), which—despite being distilled—remains relatively weak. Its official report shows error rates of 14.3% (AIME24) and 18.5% (AIME25) on final answers alone, suggesting it may lack sufficient capability for reliably evaluating some benchmark problems. Moreover, smaller models often have limited conceptual understanding and can misjudge issues related to logic, calculation, or reasoning completeness. As a result, the verifier may inaccurately assess CoT correctness in some cases.
>
> 2. Verifier Bias:
> From our prior manual CoT inspections, we found that **base models—being only pretrained—often generate “raw” CoTs that, while sometimes containing minor local flaws, still demonstrate correct high-level reasoning and lead to correct final answers**. However, based on the verifier prompts provided, the verifier seems to apply very strict criteria and judge such CoTs wrong. In contrast, RLVR outputs tend to be more polished and syntactically aligned, potentially leading to more favorable evaluations—even when the underlying reasoning is not significantly improved. This may introduce bias in favor of RLVR under CoT-Pass@k, when the verifier lacks flexibility.
>
> 3. Trends at Larger k:
>  **Even using CoT-Pass@k, their results show that base models catch up to or nearly match RLVR at large k**. For example, on AMC23 and Minerva, RLVR is fully matched by the base at high k (Fig. 2). On Math500, the performance gap shrinks from \~22% to \~3% at k=128 and continues to close. AIME24 shows a similar narrowing (\~33% → \~20%) without saturation. This aligns well with our core observation.
>
> 4. Coding Tasks as a Robust Case:
>  **In coding tasks—where reasoning must pass unit tests—Pass@k and CoT-Pass@k effectively coincide**. There, we still find that base models outperform RLVR at large k, providing a strong, verifier-free sanity check supporting our conclusion.
>
> While we have not yet independently verified the accuracy of the verifier due to time constraints, we believe the above points offer possible explanations for the divergence in conclusions. We plan to conduct a more thorough check of this aspect in future.

---

### Official Review · Reviewer_6ryS · 2025-07-01

**Clarity:** 4
**Significance:** 3
**Originality:** 2
**Rating:** 6
**Confidence:** 4

**Summary:**

This paper investigates whether RLVF actually improves pass@k for reasoning LLMs beyond the base model.
Through extensive experiments, the authors find that across different datasets and RLVF algorithm settings, a consistent observation is that the fine-tuned model does not outperform the base model in pass@k when k is sufficiently large.
Case studies further show that the solved questions are largely a subset of those solvable by the base model.

**Questions:**

- What are the specific policy optimization parameters used during RLVF? For example, what values did you set for ppo_mini_batch_size and ppo_epochs in VeRL?
- Why does the "Perplexity Analysis" paragraph only analyze two samples from AIME?

**Ethical Concerns:**

["NO or VERY MINOR ethics concerns only"]

**Final Justification:**

This is a solid empirical contribution to the RL+LLM, reasoning, and post-training domains. While the claims are not entirely novel, prior work has only offered partial insights in limited settings. Therefore, I believe this work has the potential to make a meaningful impact on the community.

**Limitations:**

Yes

**Quality:**

3

**Strengths And Weaknesses:**

### Strengths:
- This work serves as a relatively systematic experimental survey on an important question—whether RLVF improves pass@k. Although it does not fully resolve the question nor provide theoretical explanations, the clear and detailed experimental results are likely to contribute positively to ongoing discussions in the reasoning LLM community.
- The writing is easy to follow, and the results are clearly organized, which is also aided by the accessibility of the core topic.

### Weaknesses:
- There is no theoretical explanation—only extensive experimental results. The paper includes some intuitive descriptions, such as the repeated reference to base models’ boundaries, but they remain informal.
- The experiments mainly focus on the Qwen model series.
- The evaluation is too centered on pass@k curves. I would appreciate more analyses like the one in the “Perplexity Analysis” paragraph.

---

> ### Author Rebuttal · Authors · 2025-07-31
>
> # Response
>
> Thanks for supporting the acceptance of our work. Please find our responses to your questions below.
>
> ### `W1: no theoretical explanation. The paper includes some intuitive descriptions, such as the repeated reference to base models’ boundaries, but they remain informal.`
>
>
> Thank you for the thoughtful comment. While our paper focuses on empirical findings, we provide an explanation for findings in Appendix C, summarized here:
>
> Current RLVR methods (e.g., PPO, GRPO) are **on-policy**, meaning they rely on the model to generate its own training samples. Given **the vast language space** (roughly $V^L \approx 15000^{4096}$, where $V$ is the vocab size and $L$ the avg response length), training from scratch is infeasible because the model would rarely encounter any positive reward under outcome-based reward settings.
>
> As a result, RLVR relies on the **pre-trained prior** to produce responses likely to earn positive reward. This dependence restricts exploration: most on-policy samples stay within the prior’s region, and every rewarded trajectory further reinforces this behavior through gradient updates. Soft-max sampling can occasionally generate out-of-prior trajectories, but they are exceedingly rare; when they do appear, they are usually invalid and receive negative outcome reward. This feedback loop amplifies the exploration bottleneck.
>
> Interestingly, we find that the findings, and intuitive explanations presented in our paper have been formalized into a theoretical framework in a recent work [1]. It provides theoretical support for the limitation we identify: under both KL-constrained and KL-free RLVR settings, the probability mass assigned by RLVR-trained models outside the base model’s support is theoretically bounded, and this upper bound is typically very small in practice. (Theorem 2.5, Corollary 2.7). This theory work further supports our findings, and the proof approach closely reflects the intuition we provide in Appendix C.
>
> We will clarify this explanation for our findings further in the revision.
>
> [1] The Invisible Leash: Why RLVR May Not Escape Its Origin. 2025 07.
>
> ### `W2: mainly focus on the Qwen model series.`
>
> Thank you for the comment. In addition to Qwen models, we also include  LLaMA-3.1 and  DeepSeek-R1-Distill-Qwen-14B as a starting model for RLVR in our paper.
>
> To further demonstrate the generality of our findings, we additionally conducted experiments on other base models and their RLVR-trained versions, including: Gemma-2-2B [1], Mimo-7B [2], Tulu-3-70B [3].
>
> The results are as follows:
> ### **Gemma-2-2B on Math500**
>
> |  | pass@1 | pass@4 | pass@16 | pass@64 | pass@256 |
> | --- | --- | --- | --- | --- | --- |
> | Base | 25.8% | 39.3% | 52.4% | 64.6% | 75.0% |
> | RLVR | 28.0% | 42.1% | 54.4% | 65.2% | 73.8% |
> | Δ (Base - RLVR) | -2.2% | -2.8% | -2.0% | -0.6% | +1.2% |
>
> ### **Mimo-7B on Math500**
>
> |  | pass@1 | pass@4 | pass@16 | pass@64 | pass@256 |
> | --- | --- | --- | --- | --- | --- |
> | Base | 39.1% | 69.4% | 89.4% | 96.6% | 98.6% |
> | RL-Zero | 65.3% | 90.6% | 96.5% | 97.8% | 98.4% |
> | Δ (Base - RLVR) | -26.2% | -21.2% | -7.1% | -1.2% | +0.2% |
>
> ### **Tulu-70B on Math500**
> For Tulu-3-70B, we evaluate on the first 100 problems of MATH500 due to the high computational cost.
>
> | Model          | k=1  | k=4  | k=16 | k=32 |
> | -------------- | ---- | ---- | ---- | ---- |
> | Tulu-3-SFT*     | 58.6% | 77.2% | 88.1% | 91.0% |
> | Tulu-3-RLVR    | 64.4% | 80.9% | 87.7% | 89.0% |
> | Δ (SFT - RLVR) | -5.8% | -3.7% | +0.4% | +2.0% |
>
> *: Tulu-3-RLVR is RL-trained based on Tulu-3-SFT.
>
> **These results further support the robustness and broader applicability of our conclusions across model families.**
>
> [1] Gemma 2: Improving Open Language Models at a Practical Size.
>
> [2] MiMo: Unlocking the Reasoning Potential of Language Model -- From Pretraining to Posttraining
>
> [3] Tulu 3: Pushing Frontiers in Open Language Model Post-Training
>
> ### `W3: The evaluation is too centered on pass@k curves. I would appreciate more analyses like the one in the “Perplexity Analysis” paragraph.`
>
> We appreciate your recognition of the perplexity analyses beyond pass@k. In our paper, we also include **accuracy distributions, reasoning coverage, and comparisons with distilled models**, intended to provide a more comprehensive view of RLVR’s capabilities and limitations.
>
> During the rebuttal, we further expanded the perplexity analysis, **adding the dynamics of perplexity throughout training**, which will be included in the revision  (see details in our response to Reviewer KdU6’s Q4 for details if you are interested).
>
> Additionally, we plan to incorporate more plausible explanations and potential solutions for RLVR's limitations to deepen the discussion. We believe these additions will strengthen the paper beyond pass@k curves and help enrich understanding of the findings. Thank you again for your constructive suggestion.
>
> ### `Q1: What are the specific policy optimization parameters used during RLVF? For example, what values did you set for ppo_mini_batch_size and ppo_epochs in VeRL?`
>
> Thank you for the question. For broad experiments, we use publicly released checkpoints from existing RLVR works such as SimpleRLZoo, DAPO, OAT, and DeepCoder. The RL training hyperparameters for these models can be found in their respective papers.
>
> For our in-depth analysis, we trained RLVR models ourselves using the following settings:
>
> | Parameter             | Value |
> | --------------------- | ----- |
> | Rollout temperature   | 1.0   |
> | Prompt batch size     | 256   |
> | Rollout number        | 8     |
> | ppo_mini_batch_size | 256   |
> | ppo_epochs          | 1     |
>
>
> This results in 8 gradient updates per rollout step, calculated as:
> (Prompt batch size × Rollout number × PPO epochs) / PPO mini-batch size.
>
> ### `Q2: Why does the "Perplexity Analysis" paragraph only analyze two samples from AIME?`
>
> Due to space limitations, we included only two examples in the paper as representative cases—the remaining problems exhibit similar patterns. In fact, we conducted perplexity analysis on all 30 AIME problems, and in most cases, the base model assigns lower perplexity to RLVR responses than to its own, and other cases comparable.
>
> |problem id|$PPL_{base}(Y_{base})$|$PPL_{base}(Y_{RL})$|
> |----------|----------------------|---------------------|
> |0|1.133|**1.065**|
> |1|1.166|**1.155**|
> |2|1.254|**1.209**|
> |3|**1.128**|1.140|
> |4|1.117|**1.076**|
> |5|**1.092**|1.103|
> |6|1.182|**1.138**|
> |7|1.122|**1.070**|
> |8|1.176|**1.076**|
> |9|1.121|**1.107**|
> |10|1.212|**1.115**|
> |11|1.307|**1.253**|
> |12|1.122|**1.066**|
> |13|1.186|**1.173**|
> |14|**1.130**|1.145|
> |15|1.110|**1.052**|
> |16|1.201|**1.106**|
> |17|1.112|**1.090**|
> |18|1.132|**1.115**|
> |19|**1.145**|1.156|
> |20|1.163|**1.044**|
> |21|**1.256**|1.283|
> |22|1.225|**1.160**|
> |23|1.154|**1.092**|
> |24|1.085|**1.055**|
> |25|**1.225**|1.247|
> |26|1.151|**1.098**|
> |27|1.130|**1.075**|
> |28|1.171|**1.125**|
> |29|1.302|**1.278**|
> |Avg.|1.167|**1.129**|
>
>
> We will include the full set of figures in the appendix of the revision.
>
> `conclusion`
>
> We hope our responses have addressed your concerns and strengthened confidence in our paper. If you have any further questions, please feel free to reach out.

---

> ### Comment · Reviewer_6ryS · 2025-08-05
>
> Thank you for the rebuttal, appreciate your effort. In addition to those 'more experiments' stuff to support the claims, I would also like to revisit the core claim of the paper itself.
>
> I believe that similar insights have been discussed in earlier works. For example, Section 5.2.2 of DeepSeekMath [1], released in February 2024, also employed similar phrasing, such as “fundamental capabilities.” I suspect there may be other works that made similar observations even earlier, though perhaps not explored this specific point as thoroughly as in your paper.
>
> Do the authors have any comments on the potential minor novelty in the main claim?
>
> [1] Deepseekmath: Pushing the limits of mathematical reasoning in open language models. arXiv:2402.03300, 2024.

---

> ### Author Response · Authors · 2025-08-05
>
> We sincerely thank you for your time, thoughtful feedback, and continued support. Regarding the concern about potential overlap with DeepSeekMath, we would like to respectfully clarify our contribution.
>
> Since the release of DeepSeek-R1 and its demonstration of “aha moments,” RLVR has received considerable attention, with a widely held belief that it could enable LLMs to develop novel reasoning abilities and support autonomous self-improvement. While sec 5.2.2 of DeepSeekMath offered the early experiment, its analysis was limited to the pass@k curve of a single 7B instruction-tuned model, two benchmarks (GSM8k and math), and one RL method (GRPO). **We believe such a narrow setup and limited discussion is insufficient to rigorously address the broader and important question of whether current RLVR methods truly expand reasoning capability coverage. Indeed, despite Deepseek-math's earlier release, the belief in RLVR's effectiveness of expanding new capacity remained widespread in the community.**
>
> To the best of our knowledge, our paper is the first to formally pose this question and provide a large-scale, systematic evaluation across diverse settings to answer the critical question. As noted in Lines 347–352, our contributions include:
>
> - **Problem Framing**: Clearly formulating the question of whether current RLVR methods expand reasoning capability coverage, challenging a widely held belief and offering insight for the community.
> - **Breadth**: Evaluation across multiple model families (Qwen, LLaMA, Olmo, Mistral), sizes (7B–32B, including frontier models like Magistral-Medium), and RL algorithms (PPO, GRPO, RLOO, Reinforce++), covering various reasoning benchmarks.
> - **Novel Depth Analysis**: Beyond pass@k, we analyze perplexity dynamics, accuracy distributions, entropy and KL ablations, and comparisons with distilled models.
>
> As Reviewer 79qN summarized:
>
> > While the main claim has been observed already in prior literature as the authors acknowledged, this paper makes a significant contribution by providing a thorough analysis and stronger evidential basis... across a large number of training domains, models, and RL algorithms.
>
> Athough DeepSeekMath surfaced a relevant experiment early on, our work is the first to formally frame and systematically answer this important question. We will clarify this more clearly in the revision.
>
> As for the concern that “there may be other works with similar observations,” we have carefully investigated and, aside from the brief mention in Section 5.2.2 of DeepSeekMath, we are not aware of any prior work that directly and comprehensively studies the reasoning capacity coverage relationship between RLVR and base models. As acknowledged in our related work section, one rare relevant paper is the concurrent work [1], which focuses on how RLVR struggles to recover the pass@k drop introduced by SFT. However, it does not directly analyze or systematically compare base vs. RLVR reasoning capabilities as our paper does.
>
>
> We hope this helps clarify our distinction from DeepSeekMath. Thank you again for your constructive engagement.
>
> [1] Dang, Xingyu, et al. "Assessing diversity collapse in reasoning." March, 2025.

---

> > ### Comment · Reviewer_6ryS · 2025-08-06
> >
> > Thank you for the detailed follow-up clarifications and the explanations regarding the novelty. I encourage the authors to include a clearer explanation of the empirical contribution in the revised paper.
> >
> > I acknowledge that the paper proposes a verbose empirical study of the important topic. The rebuttal addresses several of my concerns. I will update the rating to 6 to reflect this.

---

> > > ### Author Response · Authors · 2025-08-06
> > >
> > > Thank you very much for your thoughtful and responsible review, as well as your support for our work. We will make sure to include a clearer explanation of the empirical contribution in the revised paper.

---

### Official Review · Reviewer_tKuY · 2025-07-03

**Clarity:** 3
**Significance:** 3
**Originality:** 3
**Rating:** 6
**Confidence:** 3

**Summary:**

This paper systematically investigates whether RLVR genuinely enhances LLM reasoning beyond base model capabilities. Using pass@k metrics across math, coding, and visual reasoning tasks, the authors find that RLVR improves sampling efficiency (pass@1) but narrows reasoning boundaries (pass@k for large k). Key finding: RLVR models only exploit reasoning paths already present in base models rather than discovering new strategies.

**Questions:**

- How do the findings extend to larger base models (e.g., 70B+ parameters)? Would the reasoning boundary limitations persist?
- Could different reward structures (e.g., process rewards vs. outcome rewards) change the conclusions about RLVR's effectiveness?

**Ethical Concerns:**

["NO or VERY MINOR ethics concerns only"]

**Final Justification:**

Rating 6.
Most of my concerns are sovled

**Limitations:**

- Evaluation scope: Limited to current RLVR methods and may not apply to future RL paradigms
- Base model selection: Results may be influenced by the specific pretrained models used

**Quality:**

4

**Strengths And Weaknesses:**

*Strengths*
- Comprehensive evaluation: Multiple model families, tasks, and RLVR algorithms tested systematically
- Novel methodology: Pass@k curves reveal hidden limitations invisible to traditional metrics
- Rigorous analysis: Perplexity analysis and manual CoT validation strongly support main claims
- Important negative result: Challenges widespread belief about RLVR's capability expansion
- Clear presentation: Well-structured with effective visualizations

*Weaknesses*
- Limited theoretical insight: Lacks explanation for why RLVR fails to expand reasoning boundaries
- Narrow scope: Findings may not generalize to future RL paradigms or different reward structures
- Missing solutions: Identifies problems clearly but provides limited improvement suggestions
- Base model dependency: Conclusions may be influenced by specific base models tested

---

> ### Author Rebuttal · Authors · 2025-07-31
>
> # Response
>
> Thanks for supporting the acceptance of our work. Please find our responses to your questions below.
>
> ### `W1: Limited theoretical insight: Lacks explanation for why RLVR fails to expand reasoning boundaries`
>
> Thank you for the thoughtful comment. While our paper focuses on empirical findings, we provide an intuitive explanation in Appendix C, summarized here:
>
> Current RLVR methods (e.g., PPO, GRPO) are **on-policy**, meaning they rely on the model to generate its own training samples. Given **the vast language space** (roughly $V^L \approx 150000^{4096}$, where $V$ is the vocab size and $L$ the avg response length), training from scratch is infeasible because the model would rarely encounter any positive reward under outcome-based reward settings.
>
> As a result, RLVR relies on the **pre-trained prior** to produce responses likely to earn positive reward. This dependence restricts exploration: most on-policy samples stay within the prior’s region, and every rewarded trajectory further reinforces this behavior through gradient updates. Soft-max sampling can occasionally generate out-of-prior trajectories, but they are exceedingly rare; when they do appear, they are usually invalid and receive negative outcome reward. This feedback loop amplifies the exploration bottleneck.
>
> Interestingly, we find that the findings, and intuitive explanations presented in our paper have been formalized into a theoretical framework in a recent work [1]. It provides theoretical support for the limitation we identify: under both KL-constrained and KL-free RLVR settings, the probability mass assigned by RLVR-trained models outside the base model’s support is theoretically bounded, and this upper bound is typically very small in practice. (Theorem 2.5, Corollary 2.7). This theory work further supports our findings, and the proof approach closely reflects the intuition we provide in Appendix C.
>
> We will clarify this explanation for our findings further in the revision.
>
> [1] The Invisible Leash: Why RLVR May Not Escape Its Origin. 2025 07.
>
> ### `W2&Q2: Narrow scope: Findings may not generalize to future RL paradigms or different reward structures;  Could different reward structures (e.g., process rewards vs. outcome rewards) change the conclusions about RLVR's effectiveness?`
>
> Thank you for the insightful question. As stated in the abstract and introduction, our study primarily aims to clarify the limitations of **current** RLVR.
>
> **By identifying what today’s RLVR can and crucially cannot achieve, we hope to motivate future RL research toward unlocking genuinely novel reasoning abilities.**
>
> Process-level rewards are indeed a promising direction for addressing current RLVR's limitation: in principle, assigning credit to intermediate reasoning steps could help alleviate the exploration inefficiency. However, current process rewards tend to be noisy and vulnerable to hacking, which is why most practical implementations still rely on outcome-based rewards—where the limitations we expose remain significant.
>
> History has shown that RL can elicit new capabilities (e.g., AlphaZero in Go). However, RLVR for LLM faces a different set of challenges, most notably, the exponentially larger action space and the strong influence of the pretrained prior, as discussed in detail in Appendix C. We believe that with further advances such as improved process rewards, RL may eventually overcome these limitations (we elaborate on potential directions in the next question). **However, before that, clearly identifying the current gap, as our paper does, is a critical first step toward that goal.**
>
> ### `W3: Missing solutions: Identifies problems clearly but provides limited improvement suggestions;`
>
> Thank you for the comment. Pinpointing the problem is only the first step; finding viable remedies is equally important for advancing LLM reasoning models. We see several promising research directions:
> 1. **Finer-grained reward structures**: step-wise rewards guide intermediate reasoning and reduce exploration bottlenecks.
> 2. **Improved exploration**: Instead of naive softmax sampling, introduce structured or hierarchical search to enhance exploration efficiency.
> 3. **Better long-horizon credit assignment**: Use techniques to propagate reward more effectively over long CoT chains and enabling the model to assign credit to crucial intermediate steps instead the whole response
> 4. **Scaling up RL training**: Match RLVR compute and data scale to that of pre-training
> 5. **Multi-turn tool use & external knowledge**: Allow the agent to interact with tools or retrieve external facts, broadening the reasoning space beyond single-pass generation
>
> While Appendix C already hints at several possible solutions, we will make these directions clearer and more concrete in the revision. Ultimately, addressing this limitation will require collective effort, and we hope our paper and the directions outlined above can help catalyze progress in this important area.
>
> ### `W4: Base model dependency: Conclusions may be influenced by specific base models tested`
>
> Thank you for the comment.  In our paper, we include  Qwen, LLaMA-3.1 and  DeepSeek-R1-Distill-Qwen-14B as a starting model for RLVR.
>
> To further demonstrate the generality of our findings, we additionally conducted experiments on other base models and their RLVR-trained versions, including: Gemma-2-2B [1], Mimo-7B [2], Tulu-3-70B [3].
>
> The results are as follows:
> ### **Gemma-2-2B on Math500**
>
> |  | pass@1 | pass@4 | pass@16 | pass@64 | pass@256 |
> | --- | --- | --- | --- | --- | --- |
> | Base | 25.8% | 39.3% | 52.4% | 64.6% | 75.0% |
> | RLVR | 28.0% | 42.1% | 54.4% | 65.2% | 73.8% |
> | Δ (Base - RLVR) | -2.2% | -2.8% | -2.0% | -0.6% | +1.2% |
>
> ### **Mimo-7B on Math500**
>
> |  | pass@1 | pass@4 | pass@16 | pass@64 | pass@256 |
> | --- | --- | --- | --- | --- | --- |
> | Base | 39.1% | 69.4% | 89.4% | 96.6% | 98.6% |
> | RL-Zero | 65.3% | 90.6% | 96.5% | 97.8% | 98.4% |
> | Δ (Base - RLVR) | -26.2% | -21.2% | -7.1% | -1.2% | +0.2% |
>
> ### **Tulu-70B on Math500**
> For Tulu-3-70B, we evaluate on the first 100 problems of MATH500 due to the high computational cost.
>
> | Model          | k=1  | k=4  | k=16 | k=32 |
> | -------------- | ---- | ---- | ---- | ---- |
> | Tulu-3-SFT*     | 58.6% | 77.2% | 88.1% | 91.0% |
> | Tulu-3-RLVR    | 64.4% | 80.9% | 87.7% | 89.0% |
> | Δ (SFT - RLVR) | -5.8% | -3.7% | +0.4% | +2.0% |
>
> *: Tulu-3-RLVR is RL-trained based on Tulu-3-SFT.
>
> **These results further support the robustness and broader applicability of our conclusions across model families.**
>
> [1] Gemma 2: Improving Open Language Models at a Practical Size.
>
> [2] MiMo: Unlocking the Reasoning Potential of Language Model -- From Pretraining to Posttraining
>
> [3] Tulu 3: Pushing Frontiers in Open Language Model Post-Training
>
> ### `Q1: How do the findings extend to larger base models (e.g., 70B+ parameters)?`
>
> Thank you for the insightful comment. We also care deeply about whether scaling up model size affects our conclusions. During the rebuttal, we conducted extensive investigations and experiments on this topic, detailed in our response to **Reviewer 79qN’s Weakness 1 (W1)**.
>
> **Our results show that the findings hold even for 70B+ models and state-of-the-art frontier models**. Due to space constraints and the absence of a global response section, we kindly refer you to that response for the full discussion.
>
> ### `conclusion`
>
> We hope our responses have addressed your concerns and strengthened confidence in our paper. If you have any further questions, please feel free to reach out.

---

> > ### Comment · Reviewer_tKuY · 2025-08-04
> >
> > Thank you for your response, my concerns have already been addresses and I have no further questions. I will keep my positive score.

---

### Official Review · Reviewer_79qN · 2025-07-08

**Clarity:** 4
**Significance:** 4
**Originality:** 3
**Rating:** 6
**Confidence:** 4

**Summary:**

The authors are interested in the following question: can LLMs fine-tuned with current RLVR approaches develop new reasoning capabilities, beyond those already present in (even if hard to elicit from) the base models? The key methodology used to examine this question is comparing the pass@k performance of base and fine-tuned models for increasingly high k values (up to 256).

Consistently, and across different models, RL algorithms and benchmarks, the authors find that base models match or outperform fine-tuned models at high k values. All of the results are on 7-32B base models (Qwen2.5, Llama-3.1), trained with standard RL algorithms (GRPO, PPO, RLOO) on a variety of reasoning benchmarks (GSM8K, MATH, Geometry3K).

The intuition provided for these results is that RLVR training narrows down the model distribution towards more-likely-to-be-rewarded reasoning paths, in the process approaching the base model's high k pass@k performance at much lower k values; but without expanding from the base model's distribution (despite the base models having the capacity to learn so, as illustrated by distillation results).

**Questions:**

1. Did you run any experiments on the models closer to the frontier, and if so, did you observe the same pattern? For example, DeepSeek-V3(-Base) vs DeepSeek-R1-Zero.

2. Similarly, do you have any experiments with SFT pre-training, does it modify the conclusions?

3. The term "reasoning capacity/boundary" seems like a bit of a misnomer; shouldn't it be solution-space/reasoning coverage or scope? Curious about the reasoning behind this term.

**Ethical Concerns:**

["NO or VERY MINOR ethics concerns only"]

**Final Justification:**

Overall, a very thorough paper and a solid contribution to understanding the effects of RLVR training.

The rating was updated from Accept to Strong Accept in rebuttal, after adding results from models closer to the frontier and fixing a number of smaller issues.

**Limitations:**

The discussion on Limitations should be improved, see the first point in the Weaknesses section.

No concerns about negative societal impact.

**Quality:**

3

**Strengths And Weaknesses:**

## Strengths

The paper's research questions is very topical and important, with significant implications for developing LLMs capable of discovery. While the main claim isn’t too surprising and has been observed already in prior literature as the authors acknowledged, this paper makes a significant contribution by providing a thorough analysis and stronger evidential basis, as well as simple & intuitive explanations. This includes demonstrating the same phenomena across a large number of training domains, models and RL algorithms; alongside other supporting experiments that strengthen the story. The results are remarkably consistent across all these variations.

I also appreciate the variety of supporting/control experiments provided, from which I'd highlight the following:
- comparing surprisal of the RL responses under base and RL policy (--> RL responses are within the base model distribution)
- comparison of models trained at different checkpoints (--> shows not improved by training longer)
- distillation from reasoning traces of a much bigger model (DeepSeek-R1) (--> shows base model’s intrinsic capacity isn’t the obstacle to learning better reasoning trajectories)
- comparison over different training GRPO hyperparameters and KL divergence (--> does not change the results)
- controlling for output entropy (--> does not change the results)

## Weaknesses

The biggest limitation of the paper is that the results are all on relatively small models (7-32B parameters) and low training data diversity (for example, mathematical reasoning models are trained only on GSM8K + MATH, coding models on LeetCode and TACO): quite far from the RLVR-trained frontier systems of today. As we've seen repeatedly in the LLM literature, both factors can significantly & suddenly change properties of models (Wei et al, 22). Given the overall thoroughness of analysis, the results are still very convincing for the small-to-mid scale RLVR pipelines (which are still important and worth publishing on!), but this limitation should be acknowledged better in the paper. As is, there is no mention of this in the Discussion section.

A couple of other points, which if addressed, would make the paper stronger:
- Random guessing explanation: low % of problems with at least one correct CoT in answers on a sample of 8-25 problems is a good sanity check, but a pretty low bar to clear. Example of something that would be more assuring: report % of trajectories containing flawed CoTs, do this for larger sample sizes using LLM-as-judge, use these measurements to estimate the bound on performance attributable to random guessing
- Over-fitting to test set: it's another alternative explanation I don't see addressed in the paper (distillation and train vs test results should indicate overfitting can't fully account for it, but this connection is not made explicit)

## Minor

- In Table 1: could you also somehow note training, not just test sets?
- Fig 6: add a sentence on the relevance of comparing to perplexity of o1 responses
- Fig 8: hard to read, perhaps show only k@1 and k@256 or $\Delta_{SE}$?
- Table 4 & 5: this would be a more effective demonstration if you just reported % overlap for a larger number of benchmarks
- Fig 16, AIME24: it looks like higher temperature actually benefits the RL model at high pass@k (in contrast to the provided explanation that T=0.6 was chosen as the best regime for both models)

---

> ### Author Rebuttal · Authors · 2025-07-31
>
> # Response
>
> Thanks for supporting the acceptance of our work. Please find our responses to your questions below.
>
> ### `W1: The biggest limitation of the paper is that results are all on relatively small models (7-32B parameters) and low training data diversity; Q1: Did you run any experiments on the models closer to the frontier?`
>
> Thank you for the insightful comment. We also care deeply about whether scaling up model size affects our conclusions.
>
> After careful investigation, we found that **only three models over 70B parameters have both base and RLVR-trained versions publicly available: DeepSeek-R1-Zero, Tulu-3-70B [1], and Magistral [2].** We summarize them in the table below.
>
> For many other large models, isolating the effect of RLVR is not feasible. For example:
> - **GPT-o1**: The base model is not accessible, and training details are unclear; the final model may not rely solely on RLVR.
> - **Qwen3-235B**: The reasoning model undergoes multiple stages, including RLVR and long-CoT SFT, impossible to isolate the effect of RLVR.
> - **LLaMA-3-70B**: no corresponding RLVR version exists.
> | Model Pair | Size | API Access | Weights Access |
> | - | - | - | - |
> | DeepSeek-V3-Base / R1-Zero | 671B | ✗ | ✓ |
> | Tulu-3-SFT / Tulu-3-RLVR | 70B | ✗ | ✓ |
> | Mistral-Medium-3 / Magistral-Medium | undisclosed* | ✓ | ✗ |
>
> *: **undisclosed size but reasoning performance close to DeepSeek-R1-671B**
>
> **For Deepseek-R1-Zero**, we have done our best to make it run with limited resources (two 8xH100 GPU nodes). However, without expert engineering optimization, the throughput remains very low, around 50 tokens/s at a maximum sequence length of 32k. Runnings on AIME24 with 1024 samplings would take approximately 4 months, which makes it currently infeasible. We are actively seeking additional resources and hope to resume the process later.
>
> Thus, we selected Tulu-3-70B and Magistral.
>
> **For Tulu-3-70B**, which we hosted locally with 8 H100 GPUs, we conducted evaluations on the first 100 subsampled problems from the MATH500 benchmark due to high computational cost. The results below are consistent with our main conclusion.
>
> ### **Tulu3 on Math500**
>
> |Model|k=1|k=4|k=16|k=32|
> |-|-|-|-|-|
> |Tulu-3-SFT|58.6%|77.2%|88.1%|91.0%|
> |Tulu-3-RLVR|64.4%|80.9%|87.7%|89.0%|
> |Δ(RLVR-SFT)|+5.8%|+3.7%|-0.4%|-2.0%|
>
> **For Magistral, one of the most advanced reasoning models**, we queried the API using a maximum context length of 40k. Again, we observed the RLVR model has great improvements at low k, but little to no gain at larger k. Specifically, at k = 1, the RLVR model solves approximately 7 more problems than the base on AIME24 and 8 more on AIME25. However, as k grows, the performance gap steadily narrows. **The trend observed in this preliminary experiment further supports that our conclusion holds even for a frontier-level reasoning model.**
> ### **Magistral on AIME24**
>
> |*k*|1|4|64|256|1024|
> |-|-|-|-|-|-|
> |Mistral-Medium-3(Base)|53.5%|69.7%|80.0%|87.6%|90.0%|
> |Magistral-Medium(RLVR)|76.6%|88.8%|90.0%|93.3%|93.3%|
> |Δ(RLVR-Base)|+23.1%|+19.1%|+10.0%|+5.7%|+3.3%|
>
>
> ### **Magistral on AIME25**
>
> |*k*|1|4|64|256|1024|
> |-|-|-|-|-|-|
> |Mistral-Medium-3(Base)|33.3%|51.5%|73.2%|81.3%|93.3%|
> |Magistral-Medium(RLVR)|62.0%|75.8%|88.7%|92.1%|93.3%|
> |Δ(RLVR-Base)|+26.7%|+24.3%|+15.5%|+10.8%|0.0%|
>
>
> We will include these results  in the revision and explicitly discuss the scalability limitation. A more comprehensive evaluation on very large models is currently constrained by resources, but this is an important direction for future work.
>
> [1] Allen Institute for AI. "Tulu 3: Pushing frontiers in open language model post-training."
>
> [2] Mistral AI. "Magistral." arXiv:2506.10910.
>
> ### `W2: Random guessing explanation: Example of something that would be more assuring: report % of trajectories containing flawed CoTs, use these measurements to estimate the bound on performance attributable to random guessing`
>
> Thanks for your suggestion. If we understand correctly, by "% of trajectories containing flawed CoTs," you’re referring to the proportion of incorrect reasoning paths across multiple samples per problem. While this is a useful metric, it serves a different purpose from our analysis.
>
> As stated in Section 2.2, our goal is to assess the **coverage of model capacity—specifically, whether the model can generate at least one correct CoT for a given problem**. This is why we manually check whether at least one correct CoT appears per problem. In contrast, measuring the percentage of flawed CoTs aligns more with **average-case evaluation**, which, while valuable, does not directly capture capacity coverage.
>
> To further address concerns about random guessing, we refer to the **coding task** in Section 3.2, where success requires passing unit tests—making random guessing highly unlikely. **The same trends hold in that setting, which supports our conclusions and helps rule out random guessing as an explanation.**
>
> ### `W3: Over-fitting to test set`
>
> Thank you for raising this point. If we understand correctly, by "overfitting to the test set," you are referring to potential test data leakage during RLVR training. We will add an explicit discussion to clarify in revision, as follows:
>
> For the RLVR checkpoints in the main experiments in Section 3, the RL training data is publicly documented and does not overlap with the evaluation benchmarks. Further, in our deep analysis (e.g., RLVR train with OmniMath data), we further ensure a clean train/test split, so there is no data leakage.
>
> If we have misunderstood your concern, please let us know—we would be happy to clarify further.
>
> ### `Q2: do you have any experiments with SFT pre-training, does it modify the conclusions?`
>
> If we understand correctly, you're asking whether our conclusions hold when RLVR is applied to models already fine-tuned via SFT or distillation. If it's yes, we have included such experiments in our paper
>
> In coding task in Section 3.2, we evaluated DeepCoder-14B, which is trained using RLVR on top of DeepSeek-R1-Distill-Qwen-14B, an SFT model based on the pretrained Qwen-2.5-14B. As shown in Figure 3 in the paper and summarized below, as k increases, base models consistently catch up to and eventually surpass RL-trained models:
>
> |pass@k|1|4|16|64|
> |-|-|-|-|-|
> |DeepSeek-R1-Distill-Qwen-14B|52.7%|63.1%|70.0%|75.6%|
> |DeepCoder-14B (RLVR)|57.2%|66.6%|71.9%|75.2%|
>
>  To further support our conclusion on SFT models, we add results on math task and will add it later in revision. Here, the RLVR model is trained on DeepSeek-R1-Distill-Qwen-1.5B (an SFT model from Qwen-2.5-1.5B). Evaluated on AIME24. Similar trends are observed.
>
> |pass@k|1|4|16|64|
> |-|-|-|-|-|
> |DeepSeek-R1-Distill-Qwen-1.5B|29.9%|52.0%|71.2%|80.4%|
> |RLVR|47.3%|66.1%|75.0%|77.0%|
>
> In summary, these results reinforce our main findings, even when RLVR is applied starting from an SFT model.
>
> ### `Q3: The term "reasoning capacity/boundary" seems like a bit of a misnomer; shouldn't it be solution-space/reasoning coverage or scope? `
>
> Thank you for pointing this out. You're right that our use of terms like reasoning capacity/boundary may have been somewhat inconsistent and potentially confusing.  **"Reasoning coverage" or "reasoning scope"** more accurately reflect the concept we intended. We will revise the terminology in the revision.
>
> ### `Minors`
>
> Thank you for your careful review and valuable suggestions. We address each point below and will incorporate the corresponding changes in the revision.
>
> - **Table 1**: We currently omit training set details due to space limitations. In the revision, we will adjust the layout to make the training data clearly identifiable for each experiment.
> - **Figure 6**: The comparison with the perplexity of o1 responses serves as a baseline, illustrating how "surprising" responses from other reasoning models (like o1) appear to the base.  We will add a sentence to clarify this in the revision.
> - **Figure 8**: The original purpose of using a folded y-axis was to highlight the ordering differences between algorithms at both k=1 and k=256.  However, we acknowledge the readability concerns of fig.8, and we will replace it with a more reader-friendly version in the revision.
> - **Figure 16 (AIME24)**:  Our choice of T=0.6 follows the evaluation protocol of DeepSeek-R1 and provides stable performance across benchmarks. We found empirically that the optimal temperature for RLVR model varies by task (e.g., higher T benefits AIME24, while lower T helps AMC23), and T=0.6 offers a reasonable overall trade-off. As shown in Figure 17, we also experimented with increasing T for RLVR. While this improves performance on AIME24, on most other benchmarks it performs comparably to RLVR at T=0.6. Even on AIME24, RLVR with higher T still lags behind the base model at large k, reinforcing our main conclusion.
> - **Tables 4 & 5**: We appreciate your suggestion and will revise the tables accordingly. Specifically, we will revise the tables to show the fraction of problems in four categories: (1) both models solve, (2) only base solves, (3) only RLVR solves, and (4) neither solves. Results are as follows:
>
> |Base|SimpleRLZoo|AIME24(k=1024)|MATH500(k=128)|
> |-|-|-|-|
> |✓|✓|63.3%|92.4%|
> |✓|✗|13.3%|3.6%|
> |✗|✓|0.0%|1.0%|
> |✗|✗|23.3%|3.0%|
>
> The results highlight key observations:
>
> The overlap in correct solutions  (type 1) is dominant. There are many cases where the base model solves a problem but RLVR fails (type 2), and very few cases where RLVR succeeds while the base does not (type 3). Even in rare type 3 cases (e.g., 1% or 5 problems in MATH500), when sampling 1024 times, the base model can solve all of them.
>
> This supports our findings that RLVR rarely solve problems the base cannot, and generally achieve lower pass@k at large k, indicating reduced coverage.
>
> ### `conclusion`
>
> We hope our responses have addressed your concerns and strengthened confidence in our paper. Should there be any further questions, feel free to reach out.

---

> > ### Author Response · Authors · 2025-08-07
> >
> > Dear Reviewer 79qN,
> >
> > As there are fewer than 48 hours remaining for the author-reviewer discussion, we kindly wanted to follow up and ask whether our responses have addressed your concerns. If there are still any issues, we would greatly appreciate the opportunity to clarify them in time.
> >
> > Thank you again for your thoughtful engagement.

---

> > ### Comment · Reviewer_79qN · 2025-08-08
> > **Response to rebuttal**
> >
> > Dear authors,
> >
> > Thank you for a very detailed rebuttal and the effort put in running the additional experiments! Based on the additional experiments (larger-scale and SFT-pretrained models) and a number of other smaller improvements, I've updated my recommendation to Strong Accept.
> >
> > ------
> >
> > Remaining points of disagreement and minor comments:
> >
> > -  in the paper, please be mindful not to refer to Magistral results as indicating performance _on frontier_ models, but indicating trend still holds _closer to frontier_ (Magistral is not reasoning SOTA, but measuring diff on SOTA models is not possible)
> > -  % of incorrect reasoning paths with correct solution still matters: if it was the case that a much larger percentage of correct solutions in base models used incorrect CoTs compared to RFT-ed models, it might indicate solutions being easier to guess than what you'd expect with a flat prior. Point taken though about tasks with unit tests!

---

> > > ### Author Response · Authors · 2025-08-08
> > >
> > > We sincerely appreciate your thoughtful review and strong support. We will revise the wording on Magistral to accurately indicate that it reflects trends closer to frontier models, not reasoning SOTA.
> > >
> > > We also agree that analyzing the percentage of correct solutions with incorrect CoTs is important. While costly to investigate, we will consider it for future work.

---

### Decision · Program_Chairs · 2025-09-17

**Decision:**

Accept (oral)

**Comment:**

The paper examines an important topic --- whether math-based RL aiming to elicit reasoning abilities improves pass@k (for high enough k) or merely pass@1. The question is intriguing, because while standard RL setups optimize pass@1 in their current formulation, pass@k is equally important in some areas (e.g., when a verifier at test time is available), it ensures sufficient exploration, and can be linked with diversity of reasoning.

The paper answers this important question negatively, i.e., pass@k does not increase, even for close to frontier models. The paper is masterfully executed, and the rebuttal essentially addresses all reviewers' concernes, as highlighted by the paper perfect score, where all reviewers strongly recommend acceptance. I am happy to do the same and strongly recommend the paper for acceptance.